# Smart 3D super-resolution microscopy reveals the architecture of the RNA scaffold in a nuclear body

Enya S. Berrevoets[1,6], Laurell F. Kessler[2,6], Ashwin Balakrishnan [2], Ellen Kazumi Okuda [3,4], Michaela Müller-McNicoll [3,5], Bernd Rieger [1] ✉, Sjoerd Stallinga [1] ✉ & Mike Heilemann [2] ✉

Small subcellular organelles orchestrate key cellular functions. How biomolecules are spatially organized within these assemblies is poorly understood. Here, we report an automated super-resolution imaging and analysis workflow that integrates confocal microscopy, morphological object screening, targeted 3D super-resolution STED microscopy and quantitative image analysis. Using this smart microscopy workflow, we target the 3D organization of *NEAT1*, an architectural RNA that constitutes the structural backbone of paraspeckles, a membraneless nuclear organelle. Using site-specific labeling, morphological sorting and particle averaging, we reconstruct the morphological space of paraspeckles along their development cycle from over 10,000 individual particles. Applying spherical harmonics analysis, we report so-far unknown heterotypes of *NEAT1* RNA organization. By integrating multi-positional labeling, we determine the coarse conformation of *NEAT1* within the organelle and show that the 3' end forms a loop-like structure at the surface of the paraspeckle. Our study reveals key structural features of paraspeckle structure and growth, as well as the molecular organization of its scaffolding RNA.

The precise molecular organization of biomolecules within larger assemblies or sub-cellular structures determines its function and is highly regulated in a cell[1]. Optical super-resolution microscopy (SRM) has positioned itself as a powerful tool to visualize the nano-scale architecture in cells[2,3].

Applying SRM to unravel the nano-scale structure of large protein complexes or small organelles is, however, still challenging. This is due to, for example, imperfect labeling and a low signal-to-noise ratio (SNR), resulting in incomplete information on a particular object. If an object has a high symmetry and is highly abundant in a cell, its nanoscale structure can nevertheless be determined by acquiring a large amount of data and subjecting it to particle averaging analysis[4,5].

A very prominent example is the nuclear pore complex (NPC), a highly-symmetric multi-protein assembly that mediates nucleocytoplasmic exchange[6]. Various optical SRM studies have contributed structural information on protein organization in NPCs[7–9] that complemented information obtained from electron microscopy[10]. A more challenging situation arises when the target object in a cell shows structural variations or heterogeneities or is part of a dynamic process. In order to obtain nano-scale structural information for such structures, this demands an even larger amount of data, as well as analysis tools to extract structural variations and to group structurally similar instances[11]. In combination with multi-target labeling, even the structural organization of complex cellular machines such as the

[1]Department of Imaging Physics, Delft University of Technology, Delft, The Netherlands. [2]Institute of Physical and Theoretical Chemistry, Goethe University, Frankfurt am Main, Germany. [3]Institute of Molecular Biosciences, Goethe University, Frankfurt am Main, Germany. [4]IMPRS on Cellular Biophysics, Frankfurt am Main, Germany. [5]Max Planck Institute for Biophysics, Frankfurt am Main, Germany. [6]These authors contributed equally: Enya S. Berrevoets, Laurell F. Kessler. ✉e-mail: b.rieger@tudelft.nl; s.stallinga@tudelft.nl; heilemann@chemie.uni-frankfurt.de

endocytotic complex become accessible[12]. To further generalize the approach to low-abundant cellular objects, correlative imaging workflows that combine fast high-throughput microscopy for cell screening with target-selective high-resolution microscopy[13] and equipped with automated selection of objects[14] can be employed. However, a super-resolution imaging and analysis workflow that is capable of reporting relevant 3D structural information on low-abundant, structurally heterogeneous sub-cellular objects remains challenging.

Here, we introduce an automated super-resolution imaging and analysis workflow that integrates large field of view (FOV) confocal microscopy, morphological object screening, targeted 3D SRM and quantitative image analysis. We developed this workflow alongside the structural study of a small and membraneless nuclear organelle, the paraspeckle, an RNA-protein condensate with a diameter of ~360 nm and a global shape ranging from spherical to ellipsoidal[15]. The long non-coding RNA *Nuclear Enriched Abundant Transcript 1* (*NEAT1*) is an architectural RNA that scaffolds paraspeckles and determines their abundance and internal organization. During transcription, the 22.7 kb *NEAT1_2* isoform interacts with 40 different paraspeckle proteins, including SFPQ, NONO and PSPC1, in a concerted manner. Through RNA-protein and protein-protein interactions *NEAT1* adopts a V-shaped structure when assembled into paraspeckles with a characteristic core-shell structure, whereby the middle of the RNA lies at the core and the 5' and 3' ends set the boundary of the particle[16,17]. In contrast, the short 3.7 kb isoform, *NEAT1_1*, overlaps the 5' end of *NEAT1_2* and is not a major component of paraspeckles and may have paraspeckle-independent roles[18]. The molar ratio of the two isoforms can vary greatly depending on the cell type. For example, *NEAT1_1* and *NEAT1_2* occur in a ratio of roughly 2:1 in U-2 OS cells, while their ratio in HeLa cells is around 1:9[19]. Paraspeckles are important mediators of the nuclear stress response through the sequestration of specific proteins and RNAs, and its composition and shape is thought to be stress specific[20,21]. To understand the shape-function relationship requires 3D structural information on these heterogeneous objects.

The nuclear location of paraspeckles demands a super-resolution method that can cover the 3D depth of a cell, which can be achieved by 3D stimulated emission depletion (STED) microscopy[22]. The low abundance of 5–20 paraspeckles per nucleus[23] and the slow imaging speed of 3D STED microscopy precludes the acquisition of a large amount of data required to extract relevant structural information. Therefore, we combined a large FOV pre-screening step with a targeted 3D STED imaging of identified objects. We first measured the global size and shape heterogeneity of paraspeckles, employing RNA-targeting fluorescence in-situ hybridization (FISH) probes that target the 3' and 5' ends of *NEAT1_2*. Using an automated particle sorting and averaging algorithm, we determined the global growth of paraspeckles in 3D. Furthermore, we developed a spherical harmonics-based image analysis approach to determine the orientation of the 3' and 5' ends within a paraspeckle, which allowed us to extract the heterogeneity in *NEAT1_2* packaging in paraspeckles. Lastly, we determined the coarse conformation of *NEAT1_2* by sequentially labeling 11 different positions along *NEAT1_2* and found a region close to its 3' end that is exposed to the outside of the particle. The resultant findings allow us to draft a model for paraspeckle growth and reveal so-far unknown structural phenotypes of RNA packaging. The developed methods are transferable to other subcellular particles and will contribute to the understanding of how the molecular architecture of these systems exerts cell biological function.

## Results

### Automated, correlative high-resolution microscopy

We developed an automated volumetric 3D SRM workflow combining large FOV confocal microscopy and target-selective STED microscopy coupled to automated post-processing and analysis. First, confocal microscopy is applied for 3D imaging of a large number of cells (Fig. 1A). Next, automated image analysis identifies objects that are then subjected to targeted small volume 3D super-resolution STED microscopy (Fig. 1B). This automated procedure enables the unsupervised acquisition of a large amount of 3D super-resolution data of target objects in regions of interest (ROI) from entire cell volumes (Fig. 1C). Finally, an automated post-processing algorithm aligns and sorts the objects in the 3D super-resolution images based on their size and shape for further analysis (Fig. 1D).

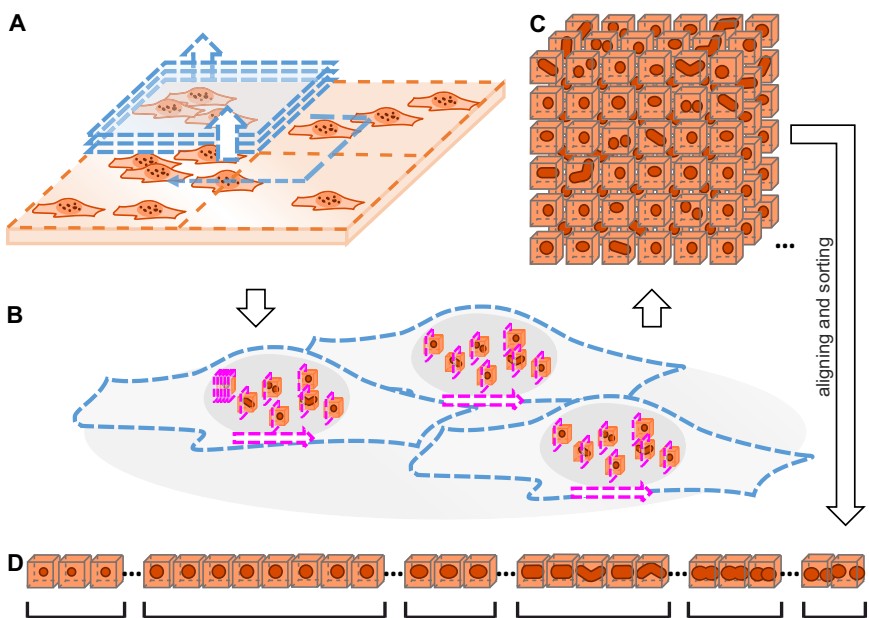

**Fig. 1 | Schematic illustration of targeted STED microscopy for low-abundant objects in 3D cells. A** Confocal screening of large FOV with multiple cells (blue) to find ROIs with possible objects. **B** Sequential volumetric 3D STED measurements on each of the ROIs (magenta). **C** A large number of 3D STED images of ROIs, each containing a single object. The screening step allowed imaging only small ROIs, thus limiting the illuminated volume and thereby reducing both the measurement time and photo bleaching. **D** The objects are aligned and sorted based on their length to facilitate downstream analysis.

## Automated 3D STED imaging of paraspeckles in cells

We applied the imaging workflow to visualize nuclear paraspeckles in fixed HeLa cells, a commonly used cell line for their structural characterization[23–26]. For this purpose, we targeted *NEAT1_2* using RNA-FISH probes extended by a single-stranded DNA handle and targeted by fluorophore-labeled locked nucleic acids (LNA) (Supplementary Table 1)[27]. Specifically, we labeled defined regions of *NEAT1_2* (e.g., the 3′ end) with several RNA-FISH probes, each equipped with the same single-stranded DNA handle and targeted by the same fluorophore-labeled LNA, in order to enhance the fluorescence signal (see "Methods") (Fig. 2A). We varied this strategy in the following three ways: (1) for a global morphological analysis of paraspeckles, we targeted both the 3′ and 5′ end of *NEAT1_2* and labeled these with the same fluorophore; (2) in order to assess the distribution of the 3′ and 5′ ends of *NEAT1_2* within paraspeckles, we targeted both the 3′ and 5′ end of *NEAT1_2* but labeled these with two different fluorophores; (3) for a coarse conformation of *NEAT1_2* in paraspeckles, we targeted the 3′ and 5′ ends of *NEAT1_2* with the same fluorophore, and added a third set of RNA-FISH probes that targets an internal position of *NEAT1_2* and is labeled with a second color. Next to enhancing the fluorescence signal recorded for specific regions of *NEAT1_2*, this approach is modular and, as only two fluorophore-labeled LNAs are needed (see "Methods"), cost efficient.

Paraspeckles are low in number and located at different positions in the nucleus. We applied the fully automated imaging workflow (Fig. 1A, B) to increase the efficiency and speed of data collection. In a first step, the full 3D volume of several pre-selected cells was recorded using low-resolution confocal microscopy over a large FOV (100 μm × 100 μm × 8 μm, isotropic pixel size of 250 nm) (Fig. 2B,Ci). Next, this imaging data was subjected to cross-correlation analysis with an ellipsoidal template representing a paraspeckle (Fig. 2Ci), returning a first estimate of the center coordinates of target objects (Fig. 2Cii). In the next step, a high-resolution confocal volume (2 μm × 2 μm × 4 μm, isotropic pixel size of 100 nm) was recorded at the position of identified objects, which was again subjected to cross-correlation analysis and reported more accurate information on the center coordinates of the object (Fig. 2Ciii). These coordinates defined the center of the ROI for super-resolution 3D STED imaging (1.1 μm × 1.1 μm × 1.1 μm, isotropic pixel size of 30 nm; resolution in Supplementary Table 2) (Fig. 2Civ). The last two steps were repeated for each individual identified object within a cell to minimize sample drift between measurements. In the case that multiple objects were identified within the high-resolution confocal volume, the coordinates were discarded and no STED measurement was performed. Supplementary Fig. 1A shows examples of accepted or rejected high-resolution confocal volumes. The ratio of accepted-to-rejected coordinates was around 4:1 (Supplementary Fig. 1B). To validate whether the automatically selected objects are indeed paraspeckles, we visualized *NEAT1_2* together with the paraspeckle marker PSPC1 (see "Methods")[28]. We performed an automated selection of coordinates using the *NEAT1_2* signal and recorded two-color STED images of *NEAT1_2* and PSPC1-GFP (Supplementary Fig. 2A). Almost 95% of the automatically selected particles showed a signal of PSPC1 (Supplementary Fig. 2B), indicating that our pipeline is robust.

## Paraspeckles are heterogeneous in size and ellipticity

For a first global analysis of the size and morphological heterogeneity of paraspeckles, we labeled the 3′ and 5′ ends of *NEAT1_2* using RNA-FISH probes targeted by two LNAs carrying the same fluorophore (Fig. 3A). The 3′ and 5′ ends of *NEAT1_2* were reported to be positioned at the surface of paraspeckles[16], and thus can serve to measure the 3D shape of the particles.

We recorded 3D STED images of 13,801 identified objects classified as bona fide paraspeckles using our automated imaging workflow (Supplementary Table 3, Supplementary Movie 1, Methods). These 3D volume images were rotated to align the major axis of the particles (i.e., the direction along which the particle is the longest) with the vertical axis of the image frame (Methods) (Supplementary Fig. 3A). The rotationally aligned images (Fig. 3B) were then sorted by their major axis lengths (Methods), binned into 250 groups of equal particle numbers (Supplementary Fig. 3B), were translationally aligned and averaged in each group (Methods) (Supplementary Fig. 3C).

For a quantitative analysis of the shape of paraspeckles, the average signal-to-center distance of the individual aligned particles was calculated along each axis to find the shell-to-center distances (Methods) (Supplementary Fig. 3D). These distances are plotted against particle number, sorted for their size as was done prior to translational alignment (Fig. 3C), and particle averaging and as histograms (Supplementary Fig. 4). The ratio between the major and the minor axis distance (Fig. 3D) represents a measure of elongation and ellipticity of the particle. Together, these two plots (Fig. 3C, D) allow us to group the shape of paraspeckles into three stages of progression: in the first stage, paraspeckles are approximately spherical, with an ellipticity of 1.14 to 1.30 (Fig. 3Ei, Supplementary Movie 2, Supplementary Table 4). In the second stage, the length of paraspeckles is increased along each axis, but at different scales and with the major axis showing the strongest increase, resulting in ellipsoidal paraspeckles with an ellipticity between 1.30 and 1.49 (Fig 3Eii, Supplementary Movie 2, Supplementary Table 4). In the third stage, the minor axis shell-to-center distance plateaued at $162 \pm 16$ nm (standard error of mean (s.e.m.)), while the intermediate axis increases shallowly between 185 and 215 nm, the major axis increases stronger to 235 and 315 nm, and the ellipticity reaches values of 2.0 (Fig 3Eiii, Supplementary Movie 2, Supplementary Table 4). 3487 out of the 13,801 imaged paraspeckles extended beyond the volume of the STED ROI (out-of-ROI; examples in Supplementary Fig. 5A). These out-of-ROI paraspeckles were analyzed separately, with a length-sorting and binning based on the intensity spread along the major axis (Methods). The averaged images reveal two additional phenotypes for elongated paraspeckles: tube-like paraspeckles with a division through the center (Fig 3Fi, Supplementary Movie 3) and closely neighboring pairs of paraspeckles (Fig 3Fii, Supplementary Movie 3). To assess the robustness of the acquisition pipeline distinguishing between paraspeckles inside and outside of an ROI, we treated HeLa cells with the proteasome inhibitor MG132, which was shown to increase the number of paraspeckles in cells and their size[29] (Supplementary Fig. 5B, C).

For all 13,801 paraspeckles, we used the fluorescence intensity as a measure for RNA quantity. To determine how the RNA quantity scales with paraspeckle volume, the intensity of images was averaged for each of the 250 sorting bins and plotted against the respective paraspeckle volume (Methods), showing a positive linear correlation (Supplementary Fig. 6, Supplementary Table 5).

## *NEAT1_2* 5′ and 3′ end frequently localize to opposite ends of the paraspeckle

Next, we sought to extract the position of the 3′ and 5′ ends of *NEAT1_2* within paraspeckles. For this purpose, we labeled the 3′ and 5′ ends of *NEAT1_2* with two spectrally orthogonal fluorophore-labeled LNAs and performed two-color 3D STED microscopy (Fig. 4A, Supplementary Movie 4, 5). We then rotationally and translationally aligned the two-color STED images and projected the 3D image intensities onto a spherical surface (Fig. 4A, left) (Methods). These surface intensity distributions were expanded into spherical harmonics (Fig. 4A, right) (Methods). The contribution of the first degree spherical harmonics to the power spectrum of the surface intensity distribution was used to measure the extent to which the 5′ and 3′ ends of *NEAT1_2* localize to the same or opposite ends of the paraspeckle, expressed as a degree of polarization $P$.

We next grouped the paraspeckles into two categories based on their ellipticity: approximately spherical paraspeckles with ellipticity

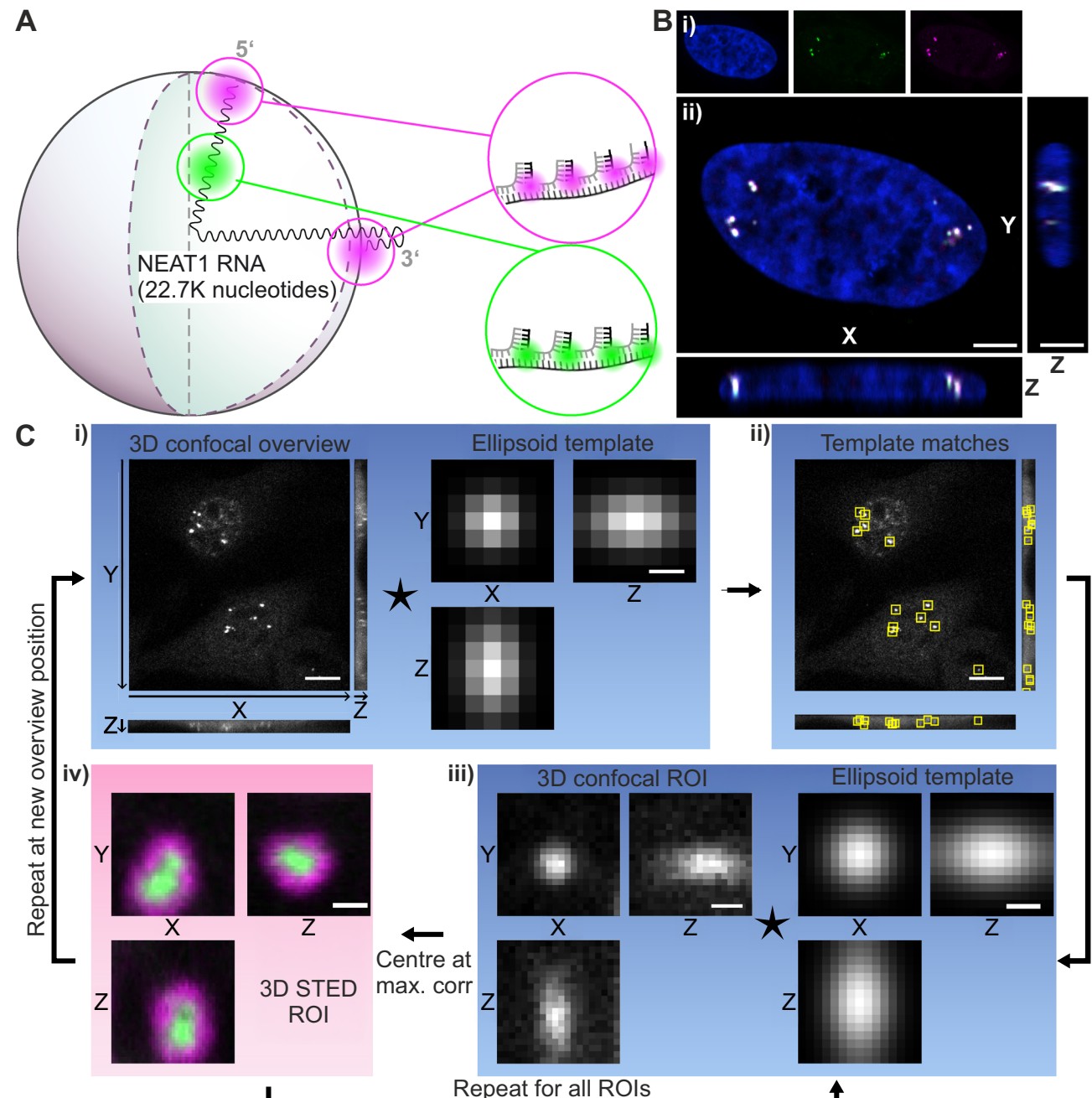

**Fig. 2 | Experimental workflow for targeted STED measurements of paraspeckles. A** RNA-FISH two color staining using two different LNA imager strands for shell (0–1K, 22–22.7K nucleotides) and core part (10–11K nucleotides) of *NEAT1_2* RNA within a paraspeckle. For shell staining, a P3-STAR 580 (Methods) imager strand was used; for core staining, a P2-STAR 635P imager strand. **B** Representative multicolor image of paraspeckles throughout the cell nucleus, measured with confocal microscopy. (i) Single channels of DAPI (blue), core (green) and shell (magenta) part of *NEAT1_2*. (ii) Ortho-sliced view of the composite. Scale bar: 4 µm. $N = 1$. **C** Automated imaging pipeline: (i) A 100 x 100 x 8 µm³ confocal overview is made with confocal resolution and isotropic 250 nm voxels. Scale bar = 10 µm (confocal overview, left); 500 nm (Ellipsoid template, right). (ii) A normalized cross-correlation is performed with an ellipsoidal template, which represents a diffraction-limited paraspeckle image, and the resulting peaks are selected as centers of regions of interest (ROIs). Scale bar = 10 µm. (iii) The first ROI is imaged again with confocal resolution, now with 100 nm voxels. The peak of another cross-correlation with an ellipsoidal template provides a more precise estimate of the paraspeckle center. Scale bar 500 nm. (iv) A 3D STED volume is acquired, centered at the found position. Shown is an exemplary paraspeckle whose shell (magenta) and core (green) region were stained with RNA-FISH. Scale bar = 300 nm. Steps ii and iii are then repeated for each ROI. Subsequently, the microscope stage is moved to a new position and the entire cycle is repeated. *The overview images in i and ii are sum projections; the single paraspeckle images in iii and iv show the central slices.*

≤1.3 and elongated paraspeckles with ellipticity >1.3. For the nearly spherical particles, the polarization ranges from 0.6% (no polarization) to 89.7% (near complete polarization), with a mean value of 44.1 ± 0.8% (s.e.m.) (Fig. 4B, Supplementary Table 6). This distribution of polarization degrees follows the same pattern as that of simulated spherical

paraspeckles with the 5' and 3' ends of *NEAT1_2* randomly distributed across the paraspeckle shell region (Supplementary Fig. 7). From the experimental data, we generated 3D volume representations showing the orientation of the 3' and 5' ends of *NEAT1_2* for low ($P < 7\%$), mid ($44\% < P < 47\%$) and high polarization ($P > 80\%$) (Fig. 4C,

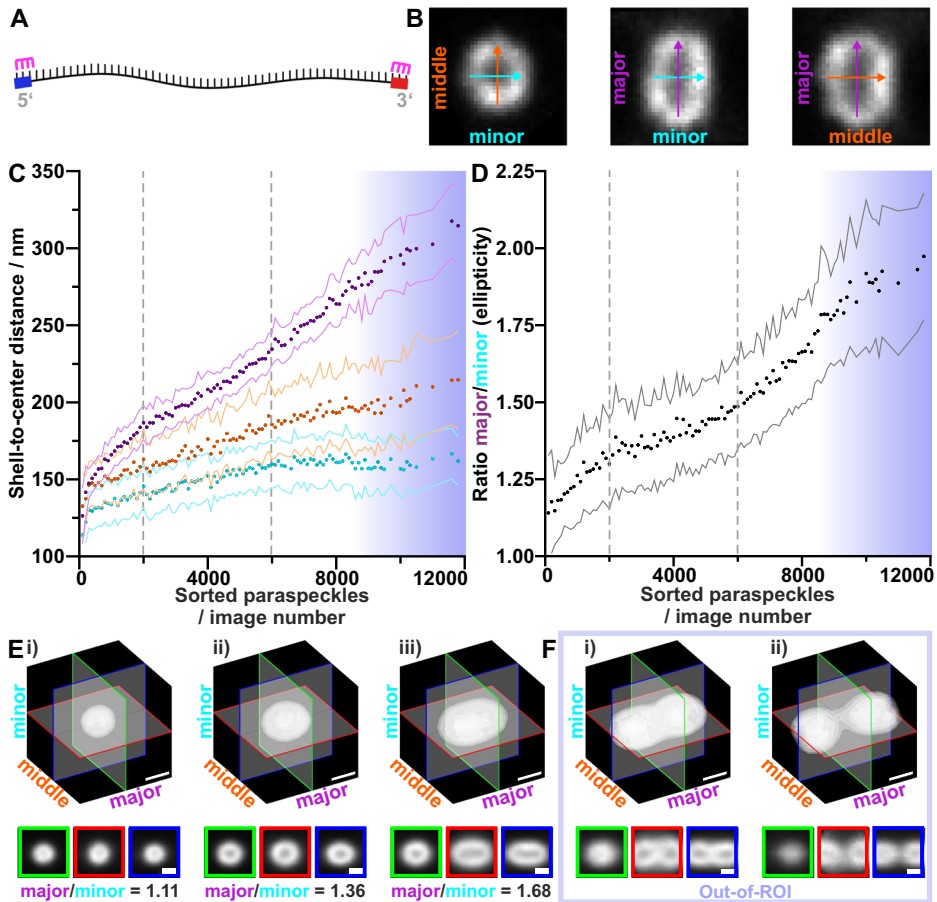

**Fig. 3 | Processing and analysis of paraspeckle volumes. A** A schematic depiction of the *NEAT1_2* RNA with the 5′ and 3′ ends indicated (blue and red, respectively). The magenta markers indicate the RNA-FISH staining on the RNA ends, marking the outside shell of paraspeckles. **B** Exemplary cross section of a paraspeckle. The arrows indicate the minor (cyan), intermediate (orange) and major axis (purple) that the particles were aligned to. The arrow lengths are based on the paraspeckle dimensions measured along each axis. **C** Measured shell-to-center distances of the paraspeckles along their minor (cyan), intermediate (orange) and major (purple) axes. 10,314 paraspeckles are included; 3487 paraspeckles were bigger than the imaged ROI (out-of-ROI) and were therefore discarded. The gray lines indicate change in slope of ellipticity and were manually selected. Blue gradient marks the out-of-ROI paraspeckles (Methods). **D** Ratio of the paraspeckles' major to minor shell-to-center distance. The points indicate the averages over 100 raw images; the corresponding standard deviation is indicated by the lines and the dotted vertical lines indicate change in slope of ellipticity and were manually selected (**C, D**). Blue gradient marks the out-of-ROI paraspeckles. **E** Exemplary 3D isosurface plots of averaged paraspeckles and corresponding cross-sections. Each average is based on 55 aligned raw images with similar major axis length. Shown are averages of images (i) 221–276 (bin 5), (ii) 3644–3698 (bin 67), (iii) 8281–8335 (bin 151). **F** Additional averages for out-of-ROI particles sorted on their covariance along the major axis. Each average is based on 55 aligned raw images with similar major axis covariance. Shown are averages of out-of-ROI paraspeckles (i) 2768–2822 (bin 51), and (ii) 3432–3487 (bin 63). Scale bar = 300 nm. *N* = 13,801 paraspeckle containing volumes from 34 independent experiments (Supplementary Table 3). Source data are provided as a Source Data file.

Supplementary Movie 4). Paraspeckles with low polarization exhibit the 3′ and 5′ ends of *NEAT1_2* scattered across the shell (Fig. 4C, left). Paraspeckles with high polarization show that the 3′ and 5′ ends of *NEAT1_2* localize at opposite sides of the particle, and thus have a dipolar structure (Fig. 4C, right). The intermediate fraction of paraspeckles shows that paraspeckles with intermediate polarization have the 3′ and 5′ ends of *NEAT1_2* distributed across the surface with a bias towards the opposite ends of the sphere (Fig. 4C, middle). 2D projections of polarized paraspeckles (*P* > 80%) show that the dipolar distribution of the 3′ and 5′ ends of *NEAT1_2* is obscured (Supplementary Fig. 8), illustrating the need for 3D high-resolution data for meaningful interpretation of the particle structure.

This analysis was repeated for the elongated paraspeckles, with polarization values ranging from 0.9% to 90.1% and a mean value of 39.8 ± 0.6% (s.e.m.) (Fig. 4D, Supplementary Table 6). The 3D volume representations show similar structural phenotypes for the orientation of the 3′ and 5′ ends in the sphere for low (*P* < 3%), mid (35% < *P* < 46%) and highly polarized (*P* > 78%) paraspeckles (Fig. 4E, Supplementary Movie 5). The distribution of the polarization degrees across the elongated paraspeckles (Fig. 4D) follows the same pattern as that of the spherical paraspeckles (Fig. 4B).

In addition to the degree of polarization, we also defined a polarization direction as a vector pointing from the 5′ dominant to the 3′ dominant region (Methods). For elongated paraspeckles (ellipticity >1.3), we can also define the angle $\alpha$ between this polarization vector and the major axis (Methods) (Fig. 4F). For non-polarized elongated paraspeckles, the polarization vector is ill-defined, and we find $\alpha = 59 \pm 3°$ (Fig. 4G, blue, Supplementary Table 7), close to the expected value for a random distribution, $\beta = 57.3$ (Methods). For polarized elongated paraspeckles, however, we find $\alpha = 71.5 \pm 1.7°$ (Fig. 4G, red, Supplementary Table 7), exceeding the random angle. This indicates a non-random relationship between the polarization vector and the elongation of the paraspeckle, with a tendency towards perpendicular orientations. For approximately spherical paraspeckles (ellipticity ≤1.3), the major axis is ill-defined, and we indeed find near-random angles $\alpha = 57 \pm 3°$ for both polarized and non-polarized paraspeckles (Supplementary Fig. 9, Supplementary Table 7). To summarize, elongated paraspeckles with polarized 3′ and 5′ ends are more

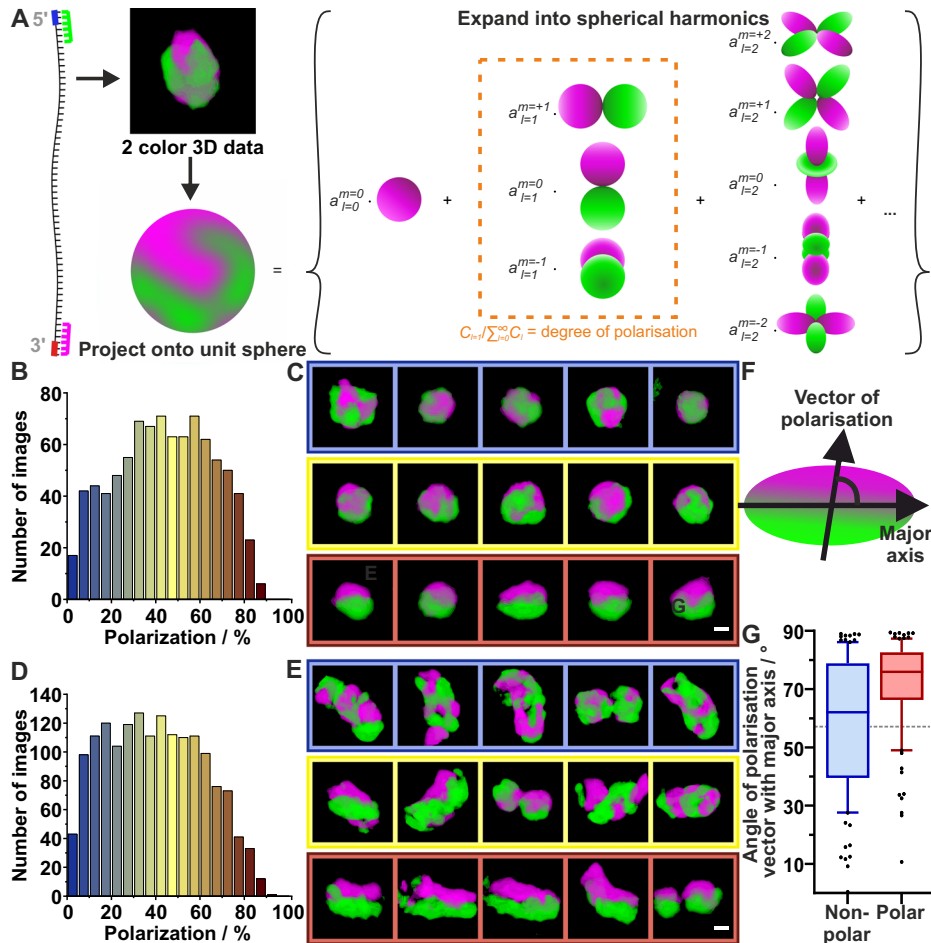

**Fig. 4 | Polarization distribution in spherical and ellipsoidal paraspeckles.**
**A** Schematic of polarization analysis of a paraspeckle recorded with two-color 3D STED. The 3D image intensity is projected onto a spherical surface and subsequently expanded into spherical harmonics. The contribution of the first degree spherical harmonics ($C_{l=1}$) to the power spectrum of the surface intensity distribution was used to measure the degree of polarization. **B** Distribution of polarization degrees among spherical paraspeckles (elongation ≤1.3; $N = 887$ paraspeckle containing volumes, 3 independent experiments). **C** Representative two-color volumes of spherical paraspeckles with low polarization ($P < 7\%$, left column), moderate polarization ($44\% < P < 47\%$, middle), and high polarization ($P > 80\%$, right). The paraspeckles were rotated to align the polarization vector with the vertical axis of the image frame. Green = 5′ signal; magenta = 3′ signal. Scale bar = 200 nm. **D** Distribution of polarization degrees among ellipsoidal paraspeckles (elongation >1.3; $N = 1626$ paraspeckle containing volumes, 3 independent

experiments). **E** Representative two-color volumes of spherical paraspeckles with low polarization ($P < 3\%$, left column), moderate polarization ($35\% < P < 46\%$, middle), and high polarization ($P > 78\%$, right). Scale bar = 200 nm. **F** Graphical depiction of an ellipsoidal, polarized paraspeckle with the polarization vector, the major axis, and the angle between them indicated. **G** Angle of polarization vector relative to the major axis for ellipsoidal paraspeckles grouped into non-polar (100 paraspeckles with lowest polarization 3 independent experiments, blue) or polar (100 paraspeckles with highest polarization, 3 independent experiments, red). Random distribution equals 57°. Values below indicate parallel orientation; values above indicate orthogonal dependency. The line inside the box indicates the median, the box indicates 25th and 75th percentile and the whiskers indicate 10th and 90th percentile. The points indicate outliers. Source data are provided as a Source Data file.

likely to be polarized perpendicular to their major axis than parallel to their major axis.

**Revealing the coarse conformation of *NEAT1_2* in paraspeckles**
Next, we aimed to determine the coarse conformation of the architectural RNA *NEAT1_2* in paraspeckles. For this purpose, we labeled *NEAT1_2* at the 3′ and 5′ ends with the same fluorophore-labeled LNA, serving as reference position, and at a second position within the RNA sequence with an orthogonal fluorophore-labeled LNA, serving as internal position (Fig. 5A, B). We performed two-color super-resolution STED microscopy of the identified paraspeckles, measuring 11 internal positions in separate imaging experiments. For each of the 11 positions, between 967 and 2992 two-color STED volumes were recorded (Supplementary Table 3). The average fluorescence intensity for each internal position increased linearly with paraspeckle volume (Supplementary Fig. 10, Supplementary Table 5).

The 3D images were rotated to align the paraspeckles along their major axis. For each of the 11 datasets, the images were binned into 20 groups based on paraspeckle major axis length (Supplementary Fig. 3B) (Methods). Next, the images were translationally aligned per bin (Methods) (Supplementary Fig. 3C). We aligned the volumes of the 11 internal positions with the transformation matrix from their corresponding reference channel. To visually assess the results, averages were made of the aligned images in each bin. We show cross-sections of one of these bin averages in Fig. 5C (Supplementary Table 8) for each of the 11 internal positions. Representative raw 2D cross-sections of individual spherical paraspeckles from all 11 RNA-FISH probe sets are shown in Supplementary Fig. 11. We observe that the signal from the RNA regions near the 5′ and 3′ ends average to hollow shells (Fig. 5C); the regions closer to the middle of the RNA produce smaller shells and regions closest to the middle appear as blobs. To quantify these different shapes, we calculated the mean distance to the

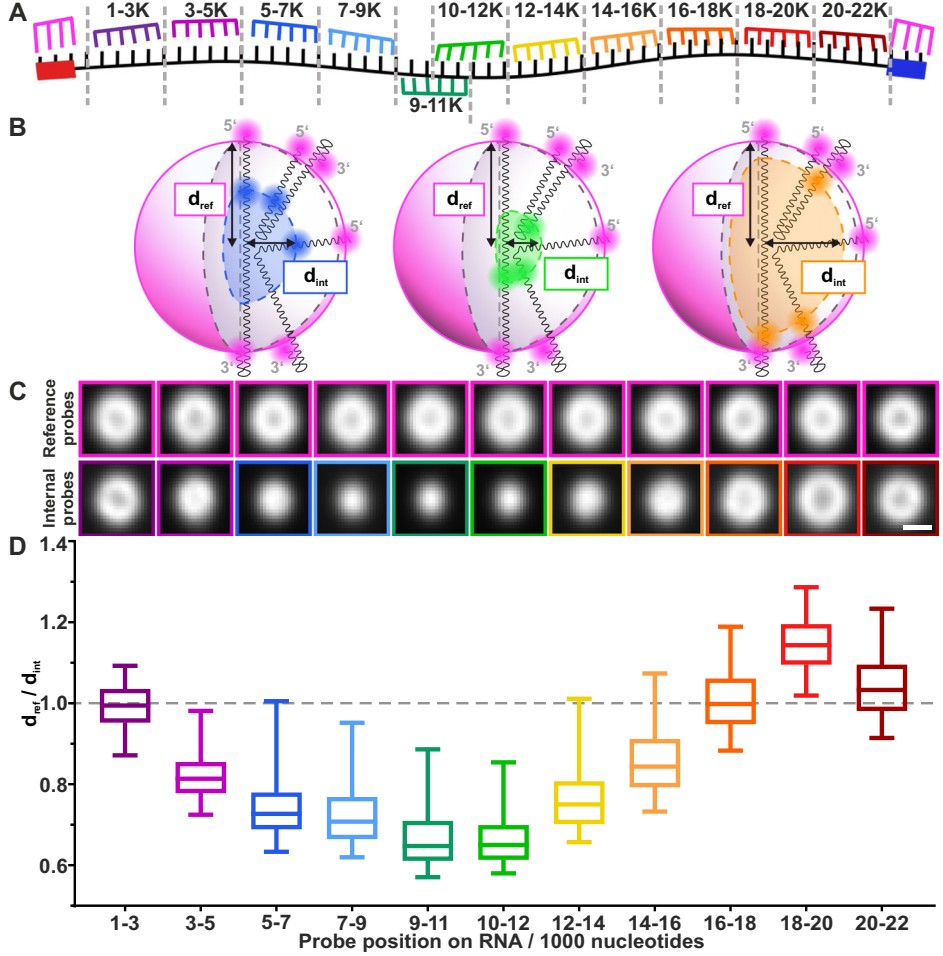

**Fig. 5 | Average localization of *NEAT1_2* regions inside the paraspeckle.**
**A** Scheme of RNA-FISH staining for 11 different 2 color measurements. In each of the 11 measurements, the reference structure is recorded by staining the RNA from 0 to 1K nt and 22K to 22.7K nt (magenta); a spectrally different staining is used to mark one of 11 regions along the rest of the RNA (rainbow color), each with a length of 2K nt, referred to as the internal probes. The internal probes binding to *NEAT1_2* in the regions of 1K to 3K and 3K to 5K can also bind to *NEAT1_1*. **B** Schematic visualization of the distance to the center (*d*) for the reference ($d_{ref}$) and internal probes ($d_{int}$). **C** Selected 2D cross-sections of averaged (*N* = 61 to 274 paraspeckles; Supplementary Table 8) spherical paraspeckles for each of the 11 two-color experiments. The top row shows the 5′ and 3′ end of *NEAT1_2*, the bottom row shows different

RNA internal probe positions. Scale bar: 300 nm. **D** The average distance of the 11 different *NEAT1_2* regions to the paraspeckle center, relative to the to-center distance of the combined 5′/3′ reference positions, as calculated from the individual images' radial intensity distribution. Lines inside the boxes indicate median, the box indicates the 25th and 75th percentile and the whiskers indicate 5th and 95th percentile (plots showing all data points are shown in Supplementary Fig. 12). *N* = 967 to 2992 paraspeckle containing volumes from up to 4 independent experiments performed per *NEAT1_2* region (Supplementary Table 3). The color-code in **C** and **D** is based on the schematic in (**A**). Source data are provided as a Source Data file.

paraspeckle center of the fluorescence signal from each RNA position (Methods) (Supplementary Fig. 3D; Fig. 5D; Supplementary Table 9). For simplification, we removed the outliers in Fig. 5D. The distribution including all data points is shown in Supplementary Fig. 12A. From nucleotides 1000 to 3000 (1–3K), near the 5′ end, to nucleotides 9–11K, the average fluorescence signal approaches the center of the paraspeckle. From nucleotides 10–12K to 16–18K, the average fluorescence signal approaches the outer boundary of the shell again. Interestingly, for nucleotides 18–20K, the fluorescence signal lies outside of the shell, before again reaching the outer boundary at nucleotides 20–22K. This indicates a loop of *NEAT1_2* that is positioned outside of the putative boundary of the paraspeckle. This pattern is conserved across different paraspeckle phenotypes: in Supplementary Fig. 12B–E, the signal distances to the paraspeckle center are depicted for spherical (ellipticity ≤1.3) and elongated (ellipticity >1.3) paraspeckles separately, showing no significant differences (Supplementary Table 9). To further validate the presence of this loop region, we recorded imaging data of cells labeled for the 3′ end (22–22.7K nt) together with the 3′

loop region (18–20K nt), and compared this data to images where the 3′ end and the 5′ end (0–1K nt) were labeled. We found a regional overlap of the 3′ end with the 3′ loop, whereas this less observed for the 3′ end and the 5′ end (Supplementary Fig. 13, Supplementary Table 10). In conclusion, by visualizing different regions of the *NEAT1_2* RNA using two-color STED imaging, we report the coarse conformation of the architectural RNA *NEAT1_2* in an average paraspeckle with varying compaction levels with the 5′ end to center RNA region exhibiting higher compaction than the center-3′ counterpart, and a small protruding loop right before the 3′ end.

## Discussion
The aim of our work was to establish a smart, automated super-resolution imaging and analysis workflow that can access 3D structural information of sparse and structurally heterogeneous sub-cellular objects spread across the entire 3D volume of a cell. In this work, we investigated the ultra-structure of nuclear paraspeckles in HeLa cells using their constituent RNA, *NEAT1_2*, as the target and establish an

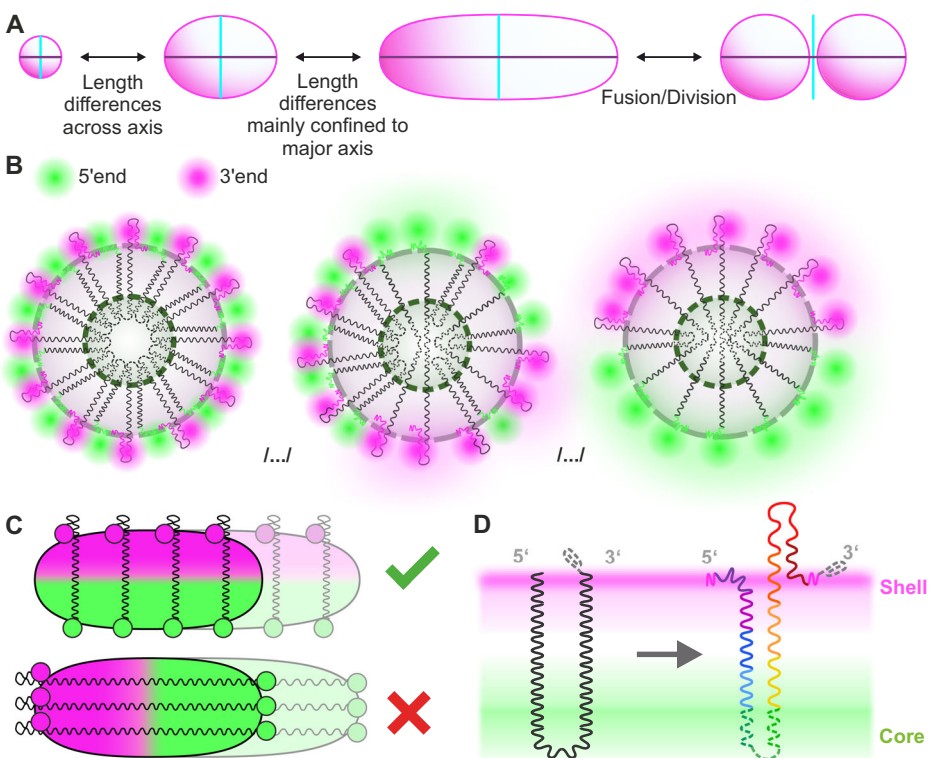

**Fig. 6 | Schematic of the proposed model of *NEAT1_2* contour along a paraspeckle. A** Schematic of the global shape of a paraspeckle and how the shape varies between different paraspeckles. Proposed model of the arrangement of how multiple RNAs could associate together in a cross section along the minor axis (**B**) and major axis (**C**) to form a paraspeckle. **D** Coarse conformation of a single *NEAT1_2* RNA in a paraspeckle Gray dotted line indicates triple helix motif at 3' end.

imaging and analysis pipeline for automated and target-attentive 3D STED imaging and subsequent analysis. We gain a better understanding of the global shape of paraspeckles and its progression in HeLa cells, and of the conformation and orientation of its major building block, the architectural RNA *NEAT1_2*.

To acquire large datasets of paraspeckles, we used a two-pronged approach. On the one hand, we developed an automated image acquisition strategy tailored to scan quickly through large volumes, identify positions of paraspeckles, and acquire volumetric STED images on these paraspeckles alone. With this volume reduction, we decreased the measurement time by three orders of magnitude and concurrently reduced the light exposure. On the other hand, we utilized a staining strategy that extends RNA-FISH by using single-stranded DNA handles which were targeted by fluorophore-labeled LNAs. The main purpose of using LNA was to gain higher SNRs due to its high affinity labeling. This combination of RNA-FISH and LNA was cost effective due to the usage of only two single-stranded LNA sequences conjugated to orthogonal fluorophores, providing a benefit over using directly fluorophore-labeled RNA-FISH probes. We optimized the target sequences along *NEAT1_2* and distinguished 11 target RNA regions, each of which covered ~2K nucleotides, as well as a 1K nucleotide region at the 5′ and 3′ ends. As the data for this manuscript were acquired in HeLa cells that show much lower levels of *NEAT1_1* as compared to *NEAT1_2* (about 1:9)[19], the sequence overlap of *NEAT1_1* and *NEAT1_2* does not affect our results. In addition, it is reported that *NEAT1_2* almost exclusively is located in paraspeckles[30], while *NEAT1_1* accumulates in paraspeckle-independent speckles (microspeckles)[18]. This might indicate that *NEAT1_1* plays a minor role in the architecture of paraspeckles. However, we point out that 5′ labeling might need consideration when working with cell lines expressing a higher fraction of *NEAT1_1*.

Our tailored acquisition and staining technique let us acquire over 16,000 volumetric paraspeckle images with high specificity, for which we developed a custom analysis workflow (Fig. 1). Analyzing the signals from the shell region of the paraspeckles, we observed that paraspeckles occur in varying sizes (shell-to-center distances varying from 52 to 485 nm) and shapes (ranging from almost spherical to elongated) (Fig. 3E, F). These results are in agreement with prior studies that reported paraspeckles exhibiting a range of sizes and shapes[17,31,32]. The morphological heterogeneity is showcased by the averaged volume images of paraspeckles (Fig. 3E, Supplementary Movie 1–3). Sorted by the major axis length, these average images demonstrate a change from small spherical paraspeckles to larger slightly ellipsoidal paraspeckles, to increasingly elongated paraspeckles. For the largest averaged paraspeckles (Figs. 3F; SV3), a division appears through the center. This suggests a fusion of or fission into multiple particles (Fig. 3Ev), a fission being the most probable from the perspective of entropy. All together, we find that most changes in paraspeckle size and shape occur along the major axis (Fig. 6A). Furthermore, our finding that the paraspeckle volume correlates linearly to the intensity measured for *NEAT1_2 RNA* hints at the possibility that the paraspeckle size and shape depends on the number of *NEAT1_2* molecules inside, while at the same time not excluding other explanations.

For all paraspeckle sizes and shapes, our volumetric datasets acquired from the 5′ and 3′ ends of *NEAT1_2* labeled with two different dyes show a seemingly random distribution of these RNA ends across the shell region of the paraspeckle (Fig. 4B, D). At one extreme of this distribution, we see the 5′ and the 3′ ends homogeneously mixed (Fig. 6B, left), which can be explained by a V-shaped *NEAT1_2* distribution along the radial cross-section supported by prior reports[33]. At the other extreme, the 5′ and 3′ ends localize to opposite ends of the paraspeckle, which is only possible if *NEAT1_2* is distributed end to end radially (Fig. 6B, right). Furthermore, for the particular phenotype of highly polarized, elongated paraspeckles, we find a high angle between the polarization direction and the direction of elongation (Fig. 4F). *NEAT1_2* must then be positioned radially along the minor axes, rather

than stretched out along the major axis (Fig. 6C). Taken together, we hypothesize that *NEAT1_2* molecules exhibit different folding patterns, from straight lines to V-shapes, and form a sheet-like arrangement (Fig. 6B) that is stacked up perpendicular to the major axis in ellipsoidal and elongated paraspeckles (Fig. 6C). This stacking arrangement is in line with the hypothesized circular skeleton of *NEAT1* perpendicular to the major axis of paraspeckles[34] albeit the difference that we observe *NEAT1_2* from straight lines to V-shapes. In addition, our hypothesis also fits the finding that paraspeckles grow upon drug (MG132) treatment[35]. Yamazaki et al. also show that paraspeckles could be considered to be made of *NEAT1_2* block copolymers, which reinforces our idea of *NEAT1_2* being arranged as units stacked up orthogonal to the major axis of the paraspeckle[35,36].

Labeling *NEAT1_2* at 11 different internal positions allowed us to determine the coarse conformation of the RNA. Our results align with previous work that reported the 3′ and 5′ end of *NEAT1_2* at the outer boundary and the mid region of *NEAT1_2* at the center of paraspeckles[15,16]. Beyond previous knowledge, we found that an RNA region near the 3′ end protrudes beyond the boundary delineated by the 5′ and 3′ ends, forming a small loop between nucleotides 16K and 22K (Figs. 5D, and 6D). This loop is conserved regardless of the shape of the paraspeckle (Supplementary Fig. 12B, C). Although the functional role of this structural conformation of *NEAT1_2* remains elusive, Van Nostrand et al. have reported that *NEAT1_2* sequence between ~16K and 22K are surprisingly devoid of any known binding sites for RNA-binding proteins[37]. Interestingly, the same region was shown to engage in long-range RNA:RNA interactions[34], suggesting that folding in this region might contribute to the formation of the external loop. Folding the 3′ end back inwards could protect *NEAT1_2* RNA from degradation in addition to the well-known triple helix motif at the 3′ end of paraspeckles (nucleotides 22,651–22,743)[38,39]. The loop might also be available for interactions other than binding to RNA binding proteins or for the formation of larger paraspeckle clusters. Our data points towards a *NEAT1_2* arrangement where the 5′ until core part of *NEAT1_2* is more compacted and the core part until the extruding loop is less compacted. This data could prove helpful for conducting simulation and biochemical studies that could shed light onto the functional role of this observed structural conformation of *NEAT1_2*.

With our analysis of the conformation of *NEAT1_2* inside paraspeckles, we want to demonstrate our smart acquisition-analysis workflow to study sub-cellular structures. A conceptually similar approach to the automated acquisition was taken by Mol and Vlijm, who implemented it to a different target[40]. The main difference between their approach and ours is that they performed 2D imaging, for which a more straightforward intensity threshold-based selection sufficed for the automated ROI detection, while the 3D volumetric imaging presented here demands for refined analysis methods. Our method also reduces the light exposure, which is also achieved with Dynamic Intensity Minimum (DyMIN) STED by scanning the depletion laser only where fluorophores are present[41]. For our purposes, this method lacks specificity as the selection is based on the presence or absence of fluorophores rather than the size and shape of the target structure. To tackle large volume acquisitions with STED, Velicky et al. introduced an optical/machine learning technique named live information-optimized nanoscopy enabling saturated segmentation (LIONESS) that can acquire large volumes in dense living brain tissue over time[42]. While LIONESS is powerful, our strategy is easier to implement and tailored to the problem of small, sparse and heterogeneous structures and is faster as smaller volumes are acquired. One other technique that also exemplifies the strength of automated microscopy is event-triggered STED imaging introduced by Alvelid et al., where STED imaging is performed only when certain events occur, e.g., vesicle trafficking[43]. This technique is designed to limit light exposure and is compatible with live-cell imaging. A future endeavor could involve combining different smart microscopy approaches,

which could prove useful for the application of high-throughput live-cell large volume imaging.

Using our smart acquisition-analysis method, we focused on the structural aspects of *NEAT1_2* RNA in paraspeckles. Our findings could be elaborated on by studying the effect of different stress conditions on the paraspeckles and by probing other targets within paraspeckles, such as the associated proteins. In the presented approach, the analysis used a fixed ROI size (Supplementary Fig. 5A), yet this can be extended to account for extensively long paraspeckles, especially those arising out of increased *NEAT1_2* expression upon MG132 treatment. Moving beyond paraspeckles, our method can be extended to other nuclear bodies and sparse (subcellular) targets exhibiting structural heterogeneity. By extension, this would allow for the application of our automated acquisition to studying organellar contact sites[44] in cells.

In summary, we present an automated microscopy workflow that combines high-throughput imaging, targeted 3D super-resolution imaging of identified objects, automated particle sorting and averaging, and advanced structural analysis. We applied this workflow to determine the morphological and structural features of nuclear paraspeckles directly in cells. We demonstrate how this integrative approach reports relevant structural information that is otherwise inaccessible. The imaging and analysis approach are transferable to other cellular targets.

## Methods

### Sample preparation: cell culture, BAC transfection and drug treatments

HeLa wild type (WT) cells (German Collection of Microorganisms and Cell Cultures GmbH, Germany) were grown in high glucose DMEM supplemented with 1%(v/v) GlutaMAX, 10%(v/v) FBS, 100 μ/mL penicillin, and 100 μg/mL streptomycin (all reagents from Gibco, Thermo-Fisher, Germany) under humidified conditions at 37 °C and 5% $CO_2$. All cells were passaged every 3–4 days or upon reaching 80% confluency.

For MG132 treatment and controls HeLa WT cells were cultivated on chamber slides (94.6190.802, SARSTEDT) overnight. On the following day, cells were treated with 10 μM MG132 (M7449-200UL, Sigma Aldrich) in fresh DMEM for 4 h. The same volume of DMSO was used as control (1.1 μL of DMSO for each 100 μL of medium). Cells were washed two times with phosphate buffer saline (PBS) prior fixation using 4% PFA (28908, Thermo Scientific).

Bacterial artificial chromosomes (BACs) harboring a sequence encoding for C-terminally GFP-tagged *PSPC1*[45] were isolated from *E. coli* DH10 cells using the NucleoBond™ Xtra Midi EF kit (Macherey-Nagel, Germany). WT HeLa cells were transfected with 1 μg purified BAC DNA per well in 6-well plates using jetPRIME (Polyplus, France). For stable integration of the BACs, cells were selected with 400 μg/mL Geneticin (G418, Gibco, ThermoFisher, Germany), sorted for single-cell clones with near-endogenous expression levels and expanded.

### Sample preparation: RNA FISH

For FISH staining RNA, HeLa cells were seeded on 8-well chambered coverglass (Sarstedt, Germany) that were coated with fibronectin (Sigma-Aldrich, Germany) at a density of 30,000 cells per well and were incubated overnight at 37 °C and 5% $CO_2$. Cells were fixed with 4% PFA in 1x PBS for 20 min, washed with PBS and permeabilized with 70% ethanol for at least 3 h at 4 °C. FISH was performed using Stellaris buffers (LG Biosearch Technologies, UK) following the manufacturer's protocol. Briefly, coverglasses were washed with Stellaris Wash Buffer A for 5 min at room temperature. Next cells were incubated with Stellaris Hybridization buffer containing the different FISH probes based on their respective positions (1 μmol/L, Supplementary Tables 1,3,11), placed in a humidified chamber and hybridized overnight at 37 °C protected from light. After hybridization, the coverslips were incubated with Stellaris Wash Buffer A. Cells were incubated with

Wash Buffer B for 5 min and washed three times with PBS. FISH probes were designed using Stellaris Probe Designer (v4.2, LG Biosearch Technologies, UK) (Supplementary Table 11). The single-stranded LNA were added at the right concentration (diluted in PBS with 500 mM NaCl) right before imaging (P2: 20 nM and P3: 30 nM; Supplementary Table 1).

For co-labeling of *NEAT1_2* and PSPC1-GFP, cells were FISH-stained as described above, followed by incubation with an antibody incubation buffer (Massive Photonics, Germany) for 30 min at room temperature. Cells were washed three times with PBS and incubated with an antibody incubation buffer containing FluoTag-x4 anti-GFP Abberior STAR RED nanobodies (NanoTag Biotechnologies, Germany) in a 1:100 dilution for 1 h at room temperature. After washing three times with PBS cells were post-fixated using 4% FA in PBS for 10 min. For imaging 30 nM P3-LNA (Supplementary Table 1) were diluted in PBS with 500 mM NaCl and added right before imaging.

All oligonucleotide sequences used for RNA FISH and imager strands were purchased commercially (Supplementary Table 1).

## Microscope

STED imaging was performed on an Abberior expert line microscope (Abberior Instruments, Germany) with an Olympus IX83 body (Olympus Deutschland GmbH, Germany) using a UPLXAPO 60x NA 1.42 oil immersion objective (Olympus Deutschland GmbH, Germany).

Confocal imaging was performed using a 561 nm excitation laser (2.9 μW at the back focal plane). Fluorescence was collected in the spectral range of 571 nm to 630 nm using an avalanche photodiode (APD). The images were acquired with a pinhole diameter of 1.0 AU, a pixel dwell time of 4 μs and a line accumulation of 1. A voxel size of 250 nm was used for overview images, respectively, 100 nm for the smaller ROI confocal volumes.

For 3D STED image acquisition of RNA-FISH stained cells, samples were excited with either a 561 nm or 640 nm pulsed excitation laser (4.1 μW and 3.9 μW at the back focal plane, respectively) and depleted using a 775 nm pulsed laser (212 mW and 275 mW at the back focal plane, respectively) with a 3D top-hat point spread function (PSF) and with an excitation delay of 750 ps and a 8 ns width. Fluorescence was collected in the spectral range of 571–630 nm (561 nm excitation) and 650–763 nm (640 nm excitation) using two APDs. The images were acquired with a pinhole diameter of 0.81 AU, line accumulation of 8, pixel dwell time of 10 μs and an isotropic voxel size of 30 nm.

For 3D STED image acquisition of immuno-stained PSPC1-GFP together with *NEAT1_2*, samples were excited with either a 561 or a 640 nm pulsed excitation laser (2.0 μW and 3.1 μW at the back focal plane) and depleted using a 775 nm pulsed laser (98 mW and 154 mW at the back focal plane). The voxel size was 35 nm.

For 2D STED image acquisition of RNA-FISH stained cells, samples were excited with either a 561 nm or 640 nm pulsed excitation laser (4.5 μW and 4.25 μW at the back focal plane, respectively) and depleted using a 775 nm pulsed laser (304 mW and 376 mW at the back focal plane, respectively) with a 2D donut shaped PSF and with an excitation delay of 750 ps. Fluorescence was collected in the spectral range of 571–630 nm (561 nm excitation) and 650–763 nm (640 nm excitation) using two APDs. The images were acquired with a pinhole diameter of 0.81 AU, line accumulation of 8, pixel dwell time of 10 μs and a pixel size of 25 nm.

## Imaging automation

The microscope hardware is accessed through the dedicated software Imspector[46]. In the Imspector graphical user interface (GUI), the user can start the image acquisition by opening a measurement window, setting the imaging parameters for this window, and starting the measurement. In the current study, three different measurement windows were used, the parameters for which are given in section 1.2:

the confocal overview window, the smaller confocal ROI window, and the 3D STED window.

Imspector has a Python interface, SpecPy, which makes measurement control through Python scripts possible. Using SpecPy, a Python script was written that enables unsupervised imaging (Fig. 2B–E) in a method similar to that proposed by Mol and Vlijm[40]. After the user has set the desired experimental parameters for the three measurement windows, they can run the script from the Imspector GUI to start the unsupervised 3D STED acquisition of 55 ROIs per hour.

Upon running the Python code from the Imspector GUI, a 100 μm × 100 μm × 8 μm overview is made in the first window with confocal resolution and isotropic 250 nm voxels (Fig. 2B). Normalized cross-correlations[47] are performed with multiple ellipsoid templates $T$ (Fig. 2C), which represent a coarse model of diffraction-limited paraspeckles. Each of these templates represents a spherical paraspeckle with a different radius, as chosen by the user upon initialization of the acquisition. For each radius $r$, a binary template image $R$, discretized to 250 nm voxels, is made first according to

$$R(x,y,) = \begin{cases} 1 & x^2 + y^2 + z^2 \leq r^2 \\ 0 & \text{else} \end{cases} \qquad (1)$$

where $x, y, z$ run from $[-r - \text{border}]$ to $[r + \text{border}]$, with the empty border set to 500 nm. Four templates were used in the current work, with radii $r$=50, 100, 150, 200 nm. The templates $T$ are then generated by convolving the binary spheres $R$ with a Gaussian function $G$, representing the microscope's PSF:

$$T = R*G, \qquad (2)$$

$$G(x,y,z) = \exp[-\frac{x^2}{2\sigma_x^2} - \frac{y^2}{2\sigma_y^2} - \frac{z^2}{2\sigma_z^2}], \qquad (3)$$

where * indicates the convolution, $\sigma_x = \sigma_y$ are set at the lateral confocal resolution of 250 nm, and $\sigma_z$ is set to 400 nm. In the four resulting normalized cross-correlation images, for each chosen template radius, the peaks higher than threshold 0.65 are then selected as the centers for ROIs.

The first ROI is imaged in the second Imspector window, again at confocal resolution but with a 2 μm × 2 μm × 4 μm FOV and isotropic 100 nm voxels (Fig. 2D). This new image is cross-correlated with the four ellipsoid templates, now discretized to finer 100 nm voxels. The ROI is discarded if all peaks in the cross-correlation images are below a threshold of 0.7, or if there are multiple non-neighboring peaks above this threshold. If the ROI is accepted, the highest cross-correlation peak is selected as the new ROI center. In the third Imspector window, a 1.11 μm × 1.11 μm × 1.11 μm voxel STED image is then made around the new ROI center with isotropic 30 nm voxels (Fig. 2E).

This pipeline is repeated for all ROIs found in the large confocal overview. When all accepted ROIs have been imaged at STED resolution, a new large FOV overview is made at a different sample position and the full process is repeated. The experiment lasts until the number of overviews set by the user has been reached, or until the user manually aborts the experiment.

## Experiments

A total of 12 two-color imaging experiments was conducted to assess the conformation of the architectural RNA *NEAT1_2* inside paraspeckles. In the first 11, Abberior STAR 635P (STAR 635P) was used to label one of the 11 consecutive nucleotide regions along the RNA (Supplementary Table 1, Fig. 4A); Abberior STAR 580 (STAR 580) was used for the 5' and 3' ends of the RNA to provide a consistent reference.

In the 12th experiment, STAR 580 was used to label the 5′ end and STAR 635P to label the 3′ end. To perform the rotational and translational alignment, reference images were created by normalizing then summing each pair of images.

Through the automated imaging pipeline, between 967 and 2992 two-color 3D paraspeckle images were acquired for each experiment during overnight measurements.

Finally, for global shape analysis, the 5′/3′ reference images of all 11 internal probe experiments were combined in one reference image set of 13,801 images.

An additional imaging experiment was performed to validate the pipeline specificity for paraspeckles. STAR 580 was used to label the 5′ and 3′ end of *NEAT1_2* and STAR 635P was used to label PSPC1. 219 3D STED images were recorded.

### Image analysis
The image analysis was performed using MATLAB (R2023a, Math-Works, USA) with additional functions from the DIPimage toolbox.

### Analysis of PSPC1 signal
Two-color 3D STED images were averaged to 2D using a z-projection. *NEAT1_2* signal was segmented using an intensity threshold. Mean fluorescence signal in the PSPC1 averages was measured inside and outside of the corresponding *NEAT1_2* segment. The mean signal outside was subtracted from the mean signal inside. Furthermore, a 5% crosstalk of the *NEAT1_2* signal was assumed and additionally subtracted from the mean signal inside. Remaining values greater than zero were assigned as PSPC1 signals.

### Rotational alignment and sorting
We rotationally aligned and sorted the paraspeckle images before further analyzing and comparing them. The alignment was based on a principal component analysis of the reference channel. For each reference image, we calculated the 3 × 3 covariance matrix $K$ using

$$K = \sum_{k,l,m} I(x_k, y_l, z_m)[x_k, y_l, z_m] \cdot [x_k, y_l, z_m]^T, \quad (4)$$

where $x, y, z$ are the center of intensity mean corrected 3D image coordinates

$$[x, y, z] = [\hat{x}, \hat{y}, \hat{z}] - \sum_{k,l,m} I(\hat{x}_k, \hat{y}_l, \hat{z}_m)[\hat{x}_k, \hat{y}_l, \hat{z}_m], \quad (5)$$

with $\hat{x}, \hat{y}, \hat{z}$ the voxel coordinates and $I$ are the image intensity values. Variance relates to how the intensity values are spatially distributed in each dimension, with a smaller object giving lower variances. The transpose of the covariance matrix was then used as rotation matrix, for both the reference and the corresponding internal probe image to align the longest axis of the object with the vertical image axis.

### Particle averaging
We used a particle averaging approach to translationally align the rotated images. Considering the heterogeneity among the imaged paraspeckles, the images were first grouped (binned) by the paraspeckles' major axis length. This length was calculated as the distance between the 50% intensity drop-off points at either end of the image's major axis (Supplementary Fig. 3B).

The binning was performed through one of two methods: with fixed bin size or with fixed length interval. The former, where a fixed number of images was assigned to each bin, was used for the combined reference image set, comprising the reference images of all 11 internal probe experiments. This set of 13,801 images was sufficiently large to assume inter-image similarity within 250 bins of 55–56 images each. For the smaller image sets of the individual internal probe

experiments and the 5′/3′ experiments, this inter-image similarity was achieved by binning by major axis length: 20 linear intervals were set from the lowest to the highest major axis length in the set, and the images were attributed to 20 corresponding bins based on their major axis length.

For each bin, the averaging was initialized with a normalized sum of all rotationally aligned reference images in the bin, centered at the image center. Each individual image was then shifted to maximize the cross-correlation with this sum. This process was repeated once, initialized now with an updated template from the sum of the shifted images. The found translations that maximize the cross-correlations with this updated template were used to align the internal probe images as well. For both the reference and the internal probe images, the aligned images were also summed per bin to generate particle averages.

The calculation of the paraspeckle length based on the 50% drop-off points fails for paraspeckles that are imaged off-center or are larger than the imaged ROI. For these images, the drop-off points would lie beyond the image borders. As a consequence, the sorting and subsequent averaging of these images is less reliable. Images with the drop-off point at the image boundary were therefore excluded from the quantitative analyses described low. To still consider them qualitatively, an alternative sorting was performed on these out-of-ROI paraspeckles based on the eigenvalues of the covariance matrix (Eq. 4), which give the image variance along each axis. The variance along the major axis is then related to the length of the paraspeckle. While it is a more convoluted length measure than the 50% drop-off, it does allow for sorting of paraspeckles larger than the ROI. Sorting, binning, and averaging was therefore additionally performed on the out of ROI paraspeckles with this eigenvalue sorting, giving more reliable averages for larger paraspeckles.

### Radial intensity distribution
To measure the distribution of each *NEAT1_2* section within the paraspeckle, we calculated its average distance to the paraspeckle center from the individual rotationally and translationally aligned images. To include non-spherical paraspeckles, the images' major and intermediate axes were first scaled by the ratio between the length of the paraspeckle along these axes and its length along the minor axis (Supplementary Fig. 3D). Now, all paraspeckles are spherical with radius equal to the length along the minor axis. The average signal-to-center distance was then found from the radial intensity distribution $I_r$, which describes the mean intensity as a function of the distance $r$ to the image center. The radial distance $\bar{d}$ for which the radial intensity below and above $\bar{d}$ are equal was selected as the average signal-to-center distance:

$$\int_0^{\bar{d}} I_r dr = \int_{\bar{d}}^{L/2} I_r dr, \quad (6)$$

where $L$ is the width of the image and $r$ is the radius. The average intermediate and major axis signal-to-center distances were then calculated by scaling this $\bar{d}$ with the previously found ratio between the axes. Hence, three average signal-to-center distance values are found for each image: one for each axis.

For paraspeckles, the minor axis is the most well-defined and consistent[17]. Because of this, the distance along the minor axis was used when comparing the localizations of different *NEAT1_2* regions.

### Intensity vs. volume
For each paraspeckle image, we calculated the sum intensity by subtracting the background and summing the pixel values. To find the background value, the intensity median of the eight 7×7×7 pixel corners of the image was calculated. We estimated each paraspeckle volume by determining the peak image intensity value, counting the

number of pixels with at least half this intensity, and multiplying this number with the pixel volume of 30 nm × 30 nm × 30 nm. To reduce the influence of noise and shape heterogeneity in determining the relationship between the image intensity and the paraspeckle volume, the intensity and volume values were then averaged over the image bins used to align and register the individual paraspeckles.

## 5′/3′ polarization

To study the relative localizations of the 5′ and the 3′ ends of *NEAT1_2*, we performed a two-color experiment with a different label on each of the two RNA ends. The resulting images were rotationally and translationally aligned using the normalized sum of the two channels as the reference.

We combined the aligned image pairs by subtracting the normalized 3′ channel from the normalized 5′ channel. We then projected the resulting 3D intensity information, in which negative values indicate 3′ dominant regions and positive values 5′ dominant regions, onto a spherical surface. To perform this projection, we first defined and discretized a spherical cone with a solid angle of $\frac{4\pi}{512}$ and its height equal to the radius of the inscribed sphere. We used this cone to sample the image volume at 512 positions on a highly uniform Fibonacci grid[48], using linear interpolation where the cones cut voxels. The intensity function on the spherical surface was then composed by attributing the sum of the intensities inside each cone to the corresponding Fibonacci grid point on the surface. With our chosen sampling method, overlaps and gaps between the cones lead to over- and undercounting of some voxels. Summing the overcount values of all cones and comparing this sum to the analytical volume of the inscribed sphere gives us the total overlap volume of 19%; the summed undercount values give us the total gap volume of 22%. On average, this error per cone is smaller than the 3D PSF volume (Supplementary Table 2). The found intensity function on the spherical surface was expanded into spherical harmonics (Fig. 4A)[49]. The expansion of the intensity function $A(\Omega) = A(\theta, \phi)$ into spherical harmonics, with $\theta$ and $\phi$ the polar and azimuthal angles, respectively, is given by

$$A(\Omega) = \sum_{l=0}^{\infty} \sum_{m=-l}^{l} a_l^m Y_l^m(\Omega) \tag{7}$$

where $Y_l^m$ is the spherical harmonic of degree $l$ and order $m$, and $a_l^m$ is the corresponding expansion coefficient. The spherical harmonic $Y_l^m$ is given by[49]

$$Y_l^m(\theta, \varphi) = (-1)^m \sqrt{\frac{2l+1}{4\pi} \frac{(l-m)!}{(l+m)!}} P_l^m(\cos(\theta)) e^{im\varphi}, \tag{8}$$

where $P_n^m$ is the associated Legendre function. The expansion coefficients $a_l^m$ are then found using:

$$a_l^m = \sum_{\varphi=\varphi_1}^{\varphi_N} \triangle\varphi \sum_{\theta=\theta_1}^{\theta_N} \triangle\theta Y_l^m(\theta, \varphi)^* A(\theta, \varphi), \tag{9}$$

where $N$ is the number of conical sections, $\triangle\varphi$ and $\triangle\theta$ are the azimuthal and polar angular separation between the conical sections, $\varphi_1$ and $\theta_1$ are the respective azimuthal and polar angle of the first conical section and $\varphi_N$ and $\theta_N$ are the azimuthal and polar angle of the $N$th (512th) conical section. The summation reflects the discrete nature of the computational calculations.

To measure the 5′/3′ polarization, i.e., how strongly the two RNA ends localize to opposite ends of the paraspeckle, we first define the power spectrum of our signal $A(\Omega)$ using its spherical harmonics expansion:

$$\text{Power} = \int |A(\Omega)|^2 d\Omega = \int \left| \sum_{l=0}^{\infty} \sum_{m=-l}^{l} a_l^m Y_l^m(\Omega) \right|^2 d\Omega = \sum_{l=0}^{\infty} \sum_{m=-l}^{l} |a_l^m|^2, \tag{10}$$

where the last step follows from the orthonormality of spherical harmonics:

$$\int Y_{l_1}^{m_1} Y_{l_2}^{*m_2} d\Omega = \delta_{l_1 l_2} \delta_{m_1 m_2}. \tag{11}$$

The contribution $C_l$ of each degree of spherical harmonics to this power spectrum is therefore given by:

$$C_l = \sum_{m=-l}^{l} |a_l^m|^2. \tag{12}$$

We are interested in the first degree spherical harmonics, which consist of two poles with opposite signs. We used their contribution to the power spectrum to measure the degree of polarization $P$:

$$P = \frac{C_1}{\sum_{l=0}^{\infty} C_l}, \tag{13}$$

with the summed power spectrum included as a normalization factor.

The polarization analysis outlined above was repeated for 1000 simulated paraspeckles. We simulated these particles by randomly distributing $30 \pm 5$ pairs of points across a spherical surface with a radius of 180 nm. These images were blurred with a Gaussian filter with $\sigma = 110$ nm to resemble the experimental data.

For the experimental data, we also calculated the direction of polarization (polarization vector) using the expression for the spherical harmonics of the first degree. On the unit sphere, these expressions are[49]:

$$Y_1^{-1} = \sqrt{\frac{3}{8\pi}} \sin(\theta) e^{-i\varphi} = \sqrt{\frac{3}{8\pi}} (x - iy)$$

$$Y_1^0 = \sqrt{\frac{3}{4\pi}} \cos(\theta) = \sqrt{\frac{3}{4\pi}} z \tag{14}$$

$$Y_1^1 = -\sqrt{\frac{3}{8\pi}} \sin(\theta) e^{i\varphi} = -\sqrt{\frac{3}{8\pi}} (x + iy).$$

Ignoring the constant $\sqrt{\frac{3}{8\pi}}$, the spherical harmonics expansion $\Psi$ using only degree $l = 1$ becomes

$$\Psi = a_1^{-1}(x - iy) + a_1^0 \sqrt{2} z - a_1^1(x + iy) = (a_1^{-1} - a_1^1)x - i(a_1^{-1} + a_1^1)y + \sqrt{2} a_1^0 z, \tag{15}$$

giving rise to polarization vector

$$\mathbf{v} = [(a_1^{-1} - a_1^1), \, -i(a_1^{-1} + a_1^1), \, \sqrt{2} a_1^0]^T \tag{16}$$

The angle $\alpha_i$ between the polarization vector and coordinate axis $i \in \{1, 2, 3\}$ is then found from the on-axis projections using

$$\alpha_i = \arccos(|v_i|) \tag{17}$$

This angle was compared against the angle $\beta$ between the $z$-axis ($i = 3$) and the vector to a random point on a sphere. To find $\beta$, we first calculated the angle between the unit orientation vector, parametrized

by spherical coordinates, and the unit vector along the $z$-axis:

$$\arccos(\hat{\mathbf{n}} \cdot \hat{\mathbf{z}}) = \arccos([\sin(\theta)\cos(\varphi), \sin(\theta)\sin(\varphi), \cos(\theta)] \cdot [0, 0, 1])$$
$$= \arccos(\cos(\theta)) = \theta$$

(18)

This angle was averaged over the surface of a hemisphere to find the random angle $\beta$:

$$\beta = \frac{1}{2\pi} \int_0^{2\pi} \int_0^{\pi/2} \arccos(\hat{\mathbf{n}} \cdot \hat{\mathbf{z}}) \sin(\theta)\, d\theta d\varphi = \frac{1}{2\pi} \int_0^{2\pi} \int_0^{\pi/2} \theta \sin(\theta)\, d\theta d\varphi$$
$$= 1\,\text{rad} \approx 57.3^\circ.$$

(19)

## Calculation of Pearson correlation coefficient

The colocalization of the fluorescence signal in two detection channels was calculated using the Pearson correlation coefficient. The single images were preprocessed by normalizing their intensities to a [0,1] scale. For each image pair, the Pearson correlation coefficient was calculated on a pixel-by-pixel basis.

## Statistics and reproducibility

No statistical method was used to predetermine sample size, and no data were excluded from the analyses. In addition, no sample size calculation was performed. All super-resolution experiments were performed at least in triplicates or at least 1000 particles were recorded. In the case of control experiments, particle numbers were limited (~70–200). Independent samples were obtained by FISH staining cells that were seeded on different days. Statistical analyses were performed using GraphPad Prism 9 (GraphPad Software, USA). The data was checked for normality using a Shapiro–Wilk test. For non-normally distributed data sets, a Mann–Whitney U test was employed. The significance level **** corresponds to a $p$ value of <0.0001.

## Reporting summary

Further information on research design is available in the Nature Portfolio Reporting Summary linked to this article.

## Data availability

All data generated, processed and used in this study have been deposited in zenodo repository under the https://doi.org/10.5281/zenodo.14179108[50]. Source data are provided with this paper.

## Code availability

The custom codes used in the manuscript are hosted in a GitLab repository (https://gitlab.tudelft.nl/E.S.Berrevoets/paraspeckles)[51].

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

## Acknowledgements

We are grateful to Abberior Instruments for technical support. We thank Rifka Vlijm for research assistance with the microscope automatization. We thank Ina Poser (MPI-CBG) for the GFP-tagged PSPC1 BAC. This work was supported by the Netherlands Organization for Scientific Research (NWO) Gravitation programme IMAGINE! (project number: 24.005.009) (E.S.B., S.S., and B.R.); the Deutsche Forschungsgemeinschaft (grants CRC 1177, project-id: 259130777 (M.H., A.B.); INST 161/1020-1 FUGG (M.H.)); the SubCellular Architecture of LifE (SCALE) consortium funded by the Goethe University Frankfurt, Germany (M.H., L.F.K., E.K.O., M.M.M.); the GRADE Center SCALE-it (L.F.K.).

## Author contributions

M.H. and L.F.K. conceived the project. E.S.B. and L.F.K. designed the automated acquisition. E.S.B., L.F.K., S.S., and B.R. conceived the analysis pipeline, and E.S.B. wrote the software. L.F.K. prepared the samples and acquired the datasets. E.S.B. performed the image analysis. E.S.B., L.F.K., M.H., S.S., and B.R. interpreted the data. M.M.M. and E.K.O. contributed cell lines, inhibitor experiments and biological knowledge about the paraspeckles. L.F.K., E.S.B., and A.B. wrote the initial draft. L.F.K., E.S.B., A.B., M.M.M., B.R., S.S., and M.H. discussed the data and wrote the manuscript.

## Funding

## Competing interests

The authors declare no competing interests.
