## [Transparent Peer Review file · Nature Communications]

Smart 3D super-resolution microscopy reveals the architecture of the RNA scaffold in a nuclear body

Corresponding Author: Professor Mike Heilemann

Version 0:

Reviewer comments:

Reviewer #1

(Remarks to the Author)

In this manuscript, the Heilemann team developed an innovative super-resolution 3D imaging pipeline combining large field-of-view (FOV) confocal microscopy and target-selective small volume 3D stimulated emission depletion (STED) microscopy coupled to automated post-processing. This image acquisition-analysis workflow aims to achieve quantitative processing of large amounts of imaging data for molecular organization of 3D structures on low-abundant, heterogeneous subcellular objects, which is a prevailing challenge in cell biology thus is much appreciated. To demonstrate the power and value of this method, the authors explored the 3D organization of the NEAT1_2 non-coding RNA in paraspeckles, the architectural backbone of these highly heterogeneous membraneless nuclear organelle. FISH probes targeting the 3' and 5' ends of NEAT1_2 known to be present at the spherical/ellipsoidal surface, were used to determine the heterogeneity of the size and shape of paraspeckles and NEAT1_2 orientation/polarization. Moreover, FISH probes at 11 sequential internal positions along NEAT1_2 were utilized to explore coarse conformation of NEAT1_2 within paraspeckles in relation to the 5' and 3' termini on the surface. The outcome of the studies corroborates many previously reported known features of NEAT1_2 hence showcased the capability of the acquisition-analysis method. In addition, new 3D structural information of NEAT1_2 and quantitative heterogeneity of paraspeckles are provided.

However, there are substantial concerns that need to be addressed:

1) Most data involved using the FISH probe targeting the 5' end of NEAT1_2. However, the authors ignored the fact that the 3.7 kb of the 5' end of NEAT1_2 completely overlaps with NEAT1_1, which is also present in paraspeckles. Thus, the detection of 5' ends of NEAT1_1 by this FISH probe cannot be excluded. This is a major issue in determining distribution/polarization of NEAT1_2 terminals, especially considering the unknown abundance of NEAT1_1 molecules within the highly heterogeneous paraspeckles. Moreover, the authors never mentioned the noise of this 5' probe in detecting NEAT1_1, which is often more abundant and can also be present in non-paraspeckle nuclear bodies. At least NEAT1_1 needs to be eliminated and relevant experiments repeated. Without such specificity control, conclusions involving the 5' probe may not reflect the actual 3D structure of NEAT1_2 within paraspeckles.

2) Current studies of paraspeckle rely on co-labeling of NEAT1_2 and known paraspeckle proteins (e.g. NONO, PSPC1, and PSF) to ensure the identity of paraspeckles and exclude NEAT1 detection in non-paraspeckle nuclear bodies. The authors should also apply their acquisition-analysis method on particles co-labeled by NEAT1_2 FISH and known paraspeckle proteins IF and compare the heterogeneity on size and shape of paraspeckles determined by FISH of NEAT1_2 alone.

Considering the fact that the 5'-end probe detects NEAT1_1 as mentioned above, perhaps co-labeling of NEAT1_2 by the 3' terminal FISH probe with the AGGF1 protein IF, which is located at the outside rim of some paraspeckles (FASEB J. 2022 Jun;36(6):e22366), may further demonstrate NEAT1_2 orientation/polarization and application value of their acquisition-analysis method as a proof of principle.

3) Whether dynamic changes of paraspeckles can indeed be measured is a critical test of the stated value of the acquisition-analysis method. The authors must demonstrate the ability to quantitatively characterize elongated paraspeckles upon MG132 induced stress compared to controls.

4) The authors found that "the paraspeckle volume is linearly dependent on the intensity measured for NEAT1_2 RNA" and concluded that the paraspeckle size and shape depend on the number of NEAT1_2 molecules inside. However, this is only

an assumption and alternative possibilities exist. Direct evidence is required to support this conclusion, by manipulation of NEAT1_2 abundance (knockdown and increased biogenesis) as shown in many previous reports.

Specific comments

a) Among the 11 FISH probes targeting internal NEAT1_2 regions inside of paraspeckles (Fig. 5), again the probes targeting 5' end and 1-3k overlap with NEAT1_1 without demonstration of NEAT1_2 specificity.

No information is provided whether these internal probes have equal efficiency as compared to probes targeting terminal regions. In fact, in table 2, the numbers of images recorded and independent measurements for terminal probes appear to be much higher than internal probes. It is crucial to exclude the possibility that differences in calculated diameter arise from differential probe efficiency.

In addition to averaged spherical paraspeckles (Fig. 5C), perhaps representative actual images should also be presented. An explanation is needed why the averaged size of paraspeckles shown in Fig. 5C appear smaller than shown in Figure 3.

b) The authors stated that the signal of the 16-22K probe protrudes beyond the boundary delineated by the 5' and 3' ends, forming a small loop (Fig. 5D). This is an important discovery and further evidence is needed to convince the readers, e.g. clear demonstration of colocalization and co-polarization of this loop with the 3' end of NEAT1_2 and/or protrusion beyond other surface markers of paraspeckles, ideally.

c) Data in Figure 5D should be carefully reviewed. The apparent discrepancy in the numbers of data points between the top and bottom 5th percentiles (probe) needs to be explained. For example, in the "5-7" group, only 23 data points fall within the 5th percentile, while more than 34 points are located in the bottom 5th percentile.

Minor concerns:

i) Light purple color is used to indicate the major axes in shell-to-center distances (Fig. 3C)" and also the "ratio of major/minor axis (Fig. 3D)". Consider using a different color in Fig. 3D for clarity.

ii) Figure 4C and E show representative image of "44%<P<47%" and "35%<P<46%", respectively. This is a slightly narrower bin than mentioned in the main text. Justification should be provided. Since the author assumes a certain distribution of the random degree of polarization, they should be able to provide the calculated significance of measured degree of polarization. Also, please clarify whether the scale bar is applied to the entire panel in Fig. 4C, 4D, and 5C.

iii) Regarding the predicted NEAT1_2 orientation, the authors hypothesized that the 5' and 3' terminals of NEAT1_2 are distributed end to end radially from straight lines to "V" shapes. Additional possibilities could also be consistent with their observation to explain the polarization, which should be discussed.

(Remarks on code availability)

Reviewer #2

(Remarks to the Author)

In this manuscript, the authors developed an automated confocal-3D STED imaging flow and analysis tools to study the 3D structures of paraspeckles, a sub-nuclear particle thought to have important functions in mediating nuclear stress. These 360 nm particles are sparsely distributed inside nuclei. The 3D STED nanoscopy is perhaps more suitable than electron microscopy in this study because of the molecular specificity and easier 3D imaging that STED can provide. STED has a limited spatial resolution, usually between 30 to 100 nm, which means the number of pixels per particle is limited. However, spatial resolution may be enough to obtain the structural information that the authors want to acquire.

The imaging experiments were well-designed. First, they designed the automated imaging flow that allowed them to acquire over 10,000 individual particles at superresolution in 3D. In this workflow, the authors used a second confocal imaging session to carefully recenter a particle for 3D-STED imaging. Second, targeting both the 3' and 5' end of NEAT1_2 and labeling these with the same fluorophore, they performed a global morphological analysis of paraspeckles. Third, in order to assess the distribution of the 3' and 5' ends of NEAT1_2 within paraspeckles, they labeled the 3' and 5' ends with two different fluorophores. Lastly, to interrogate the course conformation of NEAT1_2, they labeled the 3' and 5' ends of NEAT1_2 with the same fluorophore and different internal locations of NEAT1_2 with a second fluorophore.

The quantitative analysis using multiple mathematical approaches is very impressive. This analysis allows the author to form a detailed picture of molecular alignments. They used the 3D ellipsoidal pattern to analyze the shape of the particles. They also used spherical harmonics to analyze the distribution of 3' and 5' ends and observe different types of distribution quantified by the polarization and polarization angle. According to the quantitative analysis, they propose the 3' end forming a loop-like structure at the surface of the paraspeckle.

Overall, this manuscript presents an excellent example for demonstrating the power of 3D STED superresolution microscopy integrated with extensive quantitative analysis to address suitable problems, achieving insights into molecular components and their structural conformation. The writing and presentation of this manuscript is outstanding. This reviewer specifically appreciates the well-thought color scheme used to present complicated data sets to ease understanding.

To further improve the quality of the manuscript, please address the following questions.

- 1) The spatial resolution. The spatial resolution of the STED microscopy cannot be well defined by settings and optics. A characterization of the spatial resolution in 3D is recommended.
- 2) Bleaching effect. The same sample is imaged multiple times. The bleaching effect could be a problem, especially in the

case of 3D imaging, where the step size is set to a very small value. Please comment on the bleaching effect that may affect the signal level and accuracy of the quantitative analysis.

3) In Figure 2 iv), please indicate the meaning of the color

4) In Paragraph 4 on Page 4, "...and plotted against particle size (Fig 3C) and as histograms (Fig S2)." Please confirm if it is the particle size in Fig. 3D.

5) In Figure 3C, the x-axis is sorted bin number? Not image number? It seems the data point is for each bin, not for each image.

6) Figure 5C, please discuss why the internal probe images show bigger particles, especially at 1-3, 18-20, and 20-22.

(Remarks on code availability)

Reviewer #3

(Remarks to the Author)

The authors' work is interesting, but I am not happy with their methodology. In particular, there are several issues in the context of using spherical harmonics.

- On page 13, you write that you projected the 3D intensity information on a spherical surface. Which projection did you use? It is not uniquely defined based on the information that you provide in the manuscript.

- Two lines above equation 7 you mention that the spherical surface is expanded into spherical harmonics, but this is not true. The intensity function on the surface is expanded, not the surface itself.

- In the line below equation 7 you need to present explicitly the equations for the spherical harmonics Y_l^m and expansion coefficients a_l^m as there exist different conventions for defining them. Without mentioning the equations that you used, the analysis cannot be reproduced.

- There are several problems with equation 8.

First, "relative weights of the first degree ... expansion coefficients" sounds a bit unspecific. For example, it does not suggest that the coefficients are squared in equation 8.

Second, I see no justification for defining the polarization as in equation 8. Here the polarization can even diverge.

Why did you not simply calculate the angular power spectrum of the intensity distribution and choose the degree $l=1$ contribution from the angular power spectrum (or even better its square root, as it is frequently done in the literature) as a value for the polarization? The angular power spectrum is widely used and well known and would be a better choice than your definition.

Third, instead of dividing by the sum of all squared expansion coefficients in equation 8 (which is wrong, if I remember correctly, if one does not ensure equal contributions of the different degrees by including appropriate prefactors), you should simply normalize the intensity distribution function before you start to expand it.

Fourth, you should have taken the absolute squares of the coefficients and not just their squares. These coefficients have complex values. This means that with your definition of equation 8 you get complex values for the polarization, which does not make sense and is not in line with what you present in the main text.

- Before equation 9 you cite reference 45. Which reference do you mean here? The list of references for the manuscript ends at reference 40.

- Did you ensure that reference 45 uses the same convention for the spherical harmonics and corresponding expansion as you? If you used not the same convention, equations 9-11 are wrong in your case.

- Equation 9 is not needed at all. Instead of calculating first a spherical harmonics expansion and using then a representation of the polarization vector in spherical harmonics, you could simply expand the known intensity function into a Cartesian expansion up to degree 1. The degree 1 contribution is the polarization vector.

- Below equation 12 you mention a Fibonacci grid. There is no point in using a Fibonacci grid here (and the restriction to the polar angle seems to make it even wrong). When you want to have an expression for the angle between the z axis and a random point on the sphere, you can simply write down the unit orientation vector in 3D parametrized by spherical coordinates, multiply it by the unit vector along the z axis (scalar product), and calculate the arccos from it to obtain the angle between the z axis and another point on the sphere. Afterwards, you just need to average this over the spherical coordinates by a surface integral with a normalizing prefactor $1/(4\pi)$.

Due to these issues I cannot recommend publication of this article in its present form.

(Remarks on code availability)

As my report mentions some issues with the analysis, the code needs to be revised accordingly.

Reviewer #4

(Remarks to the Author)

(Remarks on code availability)

The project is well-organized.

Version 1:

Reviewer comments:

Reviewer #1

(Remarks to the Author)

The authors have made extensive efforts to address the questions raised in the last round of review and indeed answered/explained many of the questions raised in the rebuttal and the text. However, there are still substantial concerns that remain and need to be further addressed.

1. The authors cited previous reports and argued that HeLa cells harbor NEAT1_1:NEAT1_2 at 1:9 ratio, thus the common 5' probe used in this manuscript "should not interfere with the results". However, these previous studies still do not exclude NEAT1_1 in paraspeckles (PSP). In fact, it is unknown whether 1:9 ratio of NEAT1_1/NEAT1_2 from total RNA on northern indeed represents the ratio of these two NEAT1 isoforms in PSP. The new data in Response Figure 1 used a probe for the NEAT1_2-specific 5-7 kb region in a polarized PSP example, which is localized more in the center relevant to the reference 5'-3' probe. Nonetheless, this still does not definitively address the 5'-3' probes are indeed from NEAT1_2. Whether NEAT1_1 abundance could corrupt the 'reference' signal, thereby compromising the fundamental interpretation of PSP outlines, is critical for NEAT1/PSP field, which should be clearly demonstrated. The ideal experiment is to eliminate NEAT1_1, or at least apply this method to U2OS cells that express abundant NEAT1_1 and show whether NEAT1_1 abundance can interfere with the analysis/conclusion. This will address whether the method is limited only to cells have negligible NEAT1_1 but could not be applied to many other cells that express abundant NEAT1_1. If this is the case, the limitation of using this method for studying paraspeckles in cells with abundant NEAT1_1, should be clearly indicated.

2. Response Fig.2/S2: The text indicates 93% whereas the rebuttal indicates 95% of PSP contains PSPC1. Such colocalization is convincing but which is correct? PSPC1 appears to be localized in the center of PSPs (panel A). What is the shell-to-center distances of PSPC1 compared to the 3'-5' probe at the surface?

3. The MG132 data is a direct test for the method. This reviewer suggests including this data in supplemental info and discusses the results relevant to the literature of PSP elongation upon exposure to MG132.

4. Response Fig. 4 nicely showed 18K to 20K probe protrudes the shell boundary. Does this protrusion exhibit co-polarization with the 3' end of NEAT1_2? If such co-polarization is not consistently observed, would it undermine the interpretation?

5. According to the comparison against simulated random distribution in response figure 5, there is no significant polarization of either 5' or 3' end of NEAT1 on paraspeckles. Nonetheless, the subsequent analysis is a major claim. It is unclear why to spend large efforts to chase this statistically insignificant phenomenon? Also, the authors should run a similar simulation on elliptical particles to address that the correlation between polarizing angle and major axis is not an artifact from categorizing algorithms.

6. Nearly all original equations are improperly rendered after the revision (P11-16). Please double check and correct.

(Remarks on code availability)

Reviewer #2

(Remarks to the Author)

The authors have addressed all my questions very well. I have no further questions.

(Remarks on code availability)

The code is published on GitHub with a well-written README file. I am glad the authors mentioned the version of DIPLIB because the current version need to be built from source code. DIPLIB 2.9 can be installed with an executable installation file. However, I was not able to run the demo because of following error message in MATLAB (ver. 2023a):

Unrecognized function or variable 'PgetLocs'.

Error in PRotation (line 43)

```
[locs, Nlocs, leg] = PgetLocs([projDir '\ headDir]);
```

Nonetheless, the documentation is very clear and useful for setting up the code.

Reviewer #3

(Remarks to the Author)

The authors have strongly improved the presentation of their methodology. The mentioned errors have been corrected. With the added descriptions the previous questions have been answered and the previous confusion has been removed. I have only a few further remarks. When these issues get solved, I no longer have any objections to publication of this work.

- Above Eq. (7), "...the image volume was first divided into 512 spherical cones of equal volume.": Using spherical cones it is not possible to divide a volume into pieces without gaps or overlaps. There exist better methods that allow to partition a volume into solid angles without gaps and overlaps. This is equivalent to partitioning the surface of a sphere into areas without gaps and overlaps. The parts might then have different volumes, which is not a problem, but requires to be taken into account by appropriate weight factors.
- Eq. (9): Introduce the symbols $\Delta\varphi$, $\Delta\theta$, φ_1 , φ_N , θ_1 , θ_N
- Above Eq. (19): replace "integrated" by "averaged" as you not only integrate, but also divide by 2π .

(Remarks on code availability)

The code contains a README file, but I did not find sufficient information there on how to install or run the code.

Reviewer #4

(Remarks to the Author)

(Remarks on code availability)

Version 2:

Reviewer comments:

Reviewer #1

(Remarks to the Author)

The previous concerns are well addressed

(Remarks on code availability)

Reviewer #3

(Remarks to the Author)

1) Above Eq. (7), "...the image volume was first divided into 512 spherical cones of equal volume.":

The authors have explained in their rebuttal why they want to keep their method of dividing the volume into cones and although it is not the best possible method I agree that choosing a better method would not change the results to a relevant extent and is therefore not necessary. However, the authors did not change their manuscript in any way that prevents other readers from raising the same criticism as I did when I read this work. In particular, the authors should mention in their manuscript the following:

- The method that is used to divide the image volume into cones
- The diameter (or solid angle) of each cone
- That there are overlaps between neighboring cones
- Some numbers that show the readers that using these cones is not a problem:
 - Give an upper bound for the percentage of the overlap volume of all cones compared to the total volume of all cones.
 - Give an upper bound for the percentage of the image volume that does not belong to any cone compared to the total image volume.

2) In the line below Eq. (9) it should read "azimuthal" instead of "azimuth".

Apart from that, the changes that I have requested in my previous report have been implemented (although the change was not highlighted in the last case).

(Remarks on code availability)

Reviewer #4

(Remarks to the Author)

(Remarks on code availability)

Response to Reviewer comments

Reviewer #1 (Remarks to the Author):

1) Most data involved using the FISH probe targeting the 5' end of *NEAT1_2*. However, the authors ignored the fact that the 3.7 kb of the 5' end of *NEAT1_2* completely overlaps with *NEAT1_1*, which is also present in paraspeckles. Thus, the detection of 5' ends of *NEAT1_1* by this FISH probe cannot be excluded. This is a major issue in determining distribution/polarization of *NEAT1_2* terminals, especially considering the unknown abundance of *NEAT1_1* molecules within the highly heterogenic paraspeckles. Moreover, the authors never mentioned the noise of this 5' probe in detecting *NEAT1_1*, which is often more abundant and can also be present in non-paraspeckle nuclear bodies. At least *NEAT1_1* needs to be eliminated and relevant experiments repeated. Without such specificity control, conclusions involving the 5' probe may not reflect the actual 3D structure of *NEAT1_2* within paraspeckles.

Response: We thank the reviewer for this important comment and apologize for not having discussed *NEAT1_1* and the influence its presence would have on our results. Indeed, the first 3.7 kb of both RNA isoforms are identical; thus, the FISH probes designed for and applied in this study will target both RNAs. We were aware of this; yet in our study design, the presence of *NEAT1_1* should not interfere with our results. We appreciate the opportunity to explain our reasoning in the following:

- 1) A study by Chujo et al. quantified *NEAT1* molecules in different cell types using an improved RNA extraction method and Northern blotting. They found that in HeLa cells, which we used in this study, *NEAT1_1* and *NEAT1_2* isoforms occur in a ratio of almost 1:9¹. This means the vast majority of detectable *NEAT1* RNAs is the long isoform *NEAT1_2*.
- 2) Another study by Li and colleagues had demonstrated that *NEAT1_1* is localizing to paraspeckle-independent foci containing *NEAT1_1*, but not *NEAT1_2*. These paraspeckle-independent foci have been dubbed microspeckles and are on average smaller than the minimal size of paraspeckles². Our automatically selecting and measuring pipeline accounts for size and should exclude almost all non-paraspeckle foci. To validate this, we performed the colocalization experiment suggested by the Reviewer. We discuss this in more detail in the later part of the answer to this comment (please see below).
- 3) We know from electron microscopy analysis using probes for the 5' end, middle and 3' end, that *NEAT1_2* RNA extends throughout the core of a paraspeckle, with 5' and 3' ends aligned at the periphery³. This arrangement was further supported by a study by West and colleagues, where super-resolution structured illumination microscopy confirmed the arrangement of both 5' and 3' ends of *NEAT1_2* at the periphery of paraspeckles⁴. Therefore our reference FISH probes also exclusively stained the shell region of paraspeckles and are thus very capable of characterizing the global shape of the paraspeckles.
- 4) Together with the core/shell distribution of *NEAT1_2* RNA West et al. reported that *NEAT1_2* adopts a V-shape inside paraspeckles, which was obtained from

measurements in FUS KO cells ⁴. This V-shape of single or bundles of *NEAT1_2* is displayed in several publications ⁵⁻⁹. According to this model we expected a homogeneously distributed signal of both *NEAT1_2* ends in the paraspeckle shell region. Surprisingly, some of the paraspeckles we analysed deviated from this model. More precisely we observed a polarized signal for FISH probes that hybridize to the 3' and 5' ends. From this we concluded that this is not explainable by sticking to a V-shaped RNA model, instead *NEAT1_2* RNAs must follow a straight line through the paraspeckle. To avoid a potential corruption of the *NEAT1_2* 5' end signal due to *NEAT1_1* signal, we also investigated the distribution of FISH probes that bind in the region from 5K to 7K of *NEAT1_2*, which is outside the overlap region but still near to the 5' end. **Response Figure 1A** shows paraspeckles with 5K to 7K staining in green together with the reference staining that binds in the paraspeckle shell. This clean *NEAT1_2* 5' end signal also shows a polarised distribution in some paraspeckles. Together with the polarised uncorrupted 3' end signal shown in **Figure 4** (main manuscript) our data demonstrate that both ends of *NEAT1_2* can occur in a polarized distribution. This finding of a subset of paraspeckles with linearly arranged *NEAT1_2* molecules is unlikely to originate from the *NEAT1_1* signal, which represents only 10% of the total *NEAT1* population in HeLa cells ¹.

To further confirm this we investigated whether a different *NEAT1_1/NEAT1_2* composition of the paraspeckles influences the polarity values. **Response Figure 1B** (see below) shows the ratio of *NEAT1* 5' end signal (which detects both isoforms) to *NEAT1_2* 3' end signal plotted against the polarity and proves that there is no dependence. All taken together we conclude that our findings on the polarity are not influenced by the potential noise of our FISH probes binding to *NEAT1_1*.

Nevertheless, we apologize for the lack of classification and have therefore revised the manuscript accordingly (**introduction p.2 §3; discussion p.6 §2**).

Response Figure 1: A Paraspeckles with “clean” *NEAT1_2* 5' end (green) together with “reference” staining (magenta). Schematic illustration of RNA FISH probes staining *NEAT1_1* and *NEAT1_2* on the left and representative two-color volumes of paraspeckles with polarised 5' end signal. Scale bar = 500 nm. **B** Plot of intensity ratios between possibly corrupted *NEAT1_2* 5' end signal and uncorrupted *NEAT1_2* 3' end signal. Data

is obtained from **Figure 4**. In each instance, a total of 100 data points were summarised. Error bars represent the standard error of the mean (SEM).

2) Current studies of paraspeckle rely on co-labeling of NEAT1_2 and known paraspeckle proteins (e.g. NONO, PSPC1, and PSF) to ensure the identity of paraspeckles and exclude NEAT1 detection in non-paraspeckle nuclear bodies. The authors should also apply their acquisition-analysis method on particles co-labeled by NEAT1_2 FISH and known paraspeckle proteins IF and compare the heterogeneity on size and shape of paraspeckles determined by FISH of NEAT1_2 alone.

Considering the fact that the 5'-end probe detects NEAT1_1 as mentioned above, perhaps co-labeling of NEAT1_2 by the 3' terminal FISH probe with the AGGF1 protein IF, which is located at the outside rim of some paraspeckles (FASEB J. 2022 Jun;36(6):e22366), may further demonstrate NEAT1_2 orientation/polarization and application value of their acquisition-analysis method as a proof of principle.

Response: We thank the reviewer for this important comment. We followed the suggestion of the reviewer and conducted additional experiments to validate our experimental approach. We measured the colocalization of the identified RNA signal with the known paraspeckle protein PSPC1 to ensure highest selection quality. For this we used HeLa cells stably expressing a PSPC1-GFP fusion protein, which we labeled with a nanobody against GFP, together with FISH probes we termed "reference" in the manuscript (binding to NEAT1_2 in the regions 0 to 1K and 22K to 22.7K nts). **Response Figure 2A** shows the resultant two colour 3D STED images. Nearly 95% of the automatically selected particles showed a signal that can be assigned to PSPC1 (**Response Figure 2B**) proving the high specificity of our pipeline. After aligning and sorting along the major axis, the curve of the elongation (**Response Figure 2C**) is similar to that in **Figure 3C**. We included this validation experiment of our automated microscope for the unsupervised measurement of paraspeckles in the manuscript (**Figure S2; results p.4 §1; discussion p.7 §3; methods p.10 §1, p.10 §4, p.11 §3, p.12 §8, p.12 §10**).

These experiments were conducted with the support of Mrs. Ellen Kazumi Okuda. Because this was a significant work package, we included Mrs. Okuda as a co-author.

We also thank the reviewer for suggesting to co-stain NEAT1_2 3' ends with a protein (AGGF1) localizing to the shell region of paraspeckles to address the specificity of the FISH probes binding near the 5' end of NEAT1_2. The polarity analysis presented in our manuscript requires a high degree of labeling of our targets. Unfortunately, classical immunostaining suffers from a degree of labeling issues¹⁰ relative to RNA-FISH. Hence, we used our FISH data presented already in our manuscript. As shown earlier in *comment 1* we used NEAT1_2 staining from 5K to 7K along with our reference staining as an alternative to staining a protein in the shell region. We believe that this analysis provides a better solution for this issue (**Response Figure 1A**).

Response Figure 2: A Examples of automatically selected *NEAT1* foci. Shown is a signal of *NEAT1* stained with “reference” probes binding to the very 3’ and 5’ end of *NEAT1_2* (magenta), immunofluorescence signal of PSPC1-GFP (green) and composites of both signals. Scale bar = 500 nm. **B** Number of automatically selected and measured particles showing overlap with PSPC1-GFP immunofluorescence signal and with no overlap. 219 particles were measured. **C** Ratio of the paraspeckles’ major to minor shell-to-center distance, obtained from paraspeckles showing overlap with PSPC1 signal. In each instance, a total of 10 data points were summarized. Error bars represent the standard error of the mean (s.e.m.).

3) Whether dynamic changes of paraspeckles can indeed be measured is a critical test of the stated value of the acquisition-analysis method. The authors must demonstrate the ability to quantitatively characterize elongated paraspeckles upon MG132 induced stress compared to controls.

Response: We thank the reviewer for this suggestion. MG132 treatment is known to increase paraspeckle size and number dramatically¹¹. HeLa cells treated with MG132 contained more and bigger paraspeckles than control cells (treated with DMSO), often accumulating together (**Response Figure 3A**). Responsively, a ~3 fold increase of *NEAT1* RNA signal per cell was detected (**Response Figure 3B**).

Our intention for this manuscript was to follow the macroscopic size and shape of round to elliptical paraspeckles with the goal to further identify microscopic features within. For this reason, our pipeline excluded volumes with paraspeckles packed too densely (example in **Response Figure 3C**) during the initial confocal screening. We also excluded paraspeckles exceeding the measured STED volume in the analysis (example in **Response Figure 3D**). By applying our pipeline to MG132 treated cells, we observed that the pipeline correctly rejects more paraspeckles that are either too densely packed or too big (**Response Figure 3E**). For

accepted paraspeckles we only find minor changes in the distribution of their axis length compared to control cells (**Response Figure 3F, G**).

To analyze elongated paraspeckles expected after MG132 treatment, the overall volume measured by STED (currently $1.1\mu\text{m} \times 1.1\mu\text{m} \times 1.1\mu\text{m}$) would have to be increased. In addition, the analysis routine would have to be extended to a model capable of treating rod shaped paraspeckles and even shapes of higher complexity. This would be beyond the scope of this manuscript.

We have revised the manuscript to better explain the acquisition and analysis pipeline (**Figure S1; results p.4 §1; Fig S5; results p.5 §1**).

Response Figure 3: Pipeline performance when utilised on MG132 treated cells. **A** Confocal measurements of HeLa cells treated with DMSO (left) and MG132 (right). Cells were treated with 10 μM of MG132 (M7449-200UL, Sigma Aldrich) or similar volume of DMSO for 4 hours, fixed and stained using RNA-FISH probes hybridising to the very 3' and 5' end of *NEAT1_2*. Scale bar = 20 μm . **B** Mean NEAT1 fluorescence signal per cell for DMSO (gray) and MG132 (light red) treated HeLa cells. N = 50. **C** Examples of confocal screened ROIs containing paraspeckles. ROIs matching with a single ellipsoidal template are accepted for a subsequent STED measurement (green). ROIs containing multiple paraspeckles are rejected (red). Scale bar = 500 nm. **D** Examples of STED volumes containing paraspeckles stained with FISH probes binding to the very 3' and 5' end of *NEAT1_2*. Volumes containing paraspeckles confined within the volume of the STED ROI were accepted and further analysed; volumes containing paraspeckles that protruded outside the STED ROI volume were rejected. Scale bar = 300 nm. **E** Fractions of automatically found paraspeckles that are measured with STED and further

analysed (green), that were measured with STED but extended beyond the volume of the STED ROI (“out of ROI”; orange) and multiple paraspeckles packed densely that were not measured with STED (red). Shown is data for DMSO (gray) and MG132 (light red) treated cells. **F** Measured shell-to-center distances of the paraspeckles along their minor, intermediate and major axes for STED volumes acquired for DMSO (black, gray and light gray) and MG132 (red, light red and very light red) treated cells. The points indicate the averages over 75 raw images; the corresponding s.e.m. is indicated by the bars. **G** Ratio of the paraspeckles’ major to minor shell-to-center distance for DMSO (black) and MG132 (red) treated cells. 1500 cells were measured each.

4) The authors found that “the paraspeckle volume is linearly dependent on the intensity measured for NEAT1_2 RNA” and concluded that the paraspeckle size and shape depend on the number of NEAT1_2 molecules inside. However, this is only an assumption and alternative possibilities exist. Direct evidence is required to support this conclusion, by manipulation of NEAT1_2 abundance (knockdown and increased biogenesis) as shown in many previous reports.

Response: We thank the reviewer for this suggestion. We measured the 3D surface of PS particles from the 5’ fluorescence signal detected with 3D STED microscopy. From this, we calculated the 3D volume of the PS sphere, and the fluorescence intensity recorded for this PS. Plotting both values for a large number of particles yielded the almost linear dependency that we show in (**Figure S3 and S6; Table 4**). We agree with the reviewer that this is a correlation, and not a causation – and softened our statement in the Discussion section accordingly (**discussion p.7 §3**).

Specific comments

a) Among the 11 FISH probes targeting internal NEAT1_2 regions inside of paraspeckles (Fig. 5), again the probes targeting 5’end and 1-3k overlap with NEAT1_1 without demonstration of NEAT1_2 specificity.

No information is provided whether these internal probes have equal efficiency as compared to probes targeting terminal regions. In fact, in table 2, the numbers of images recorded and independent measurements for terminal probes appear to be much higher than internal probes. It is crucial to exclude the possibility that differences in calculated diameter arise from differential probe efficiency.

In addition to averaged spherical paraspeckles (Fig. 5C), perhaps representative actual images should also be presented. An explanation is needed why the averaged size of paraspeckles shown in Fig. 5C appear smaller than shown in Figure 3.

Response: We thank the reviewer for these insightful comments. We addressed the presence of NEAT1_1 and NEAT1_2 in an earlier response (please see above).

Each internal probe was measured with a terminal probe as a reference. The images shown in **Table 2** (updated **Table 3**) are the pooled data of all terminal probes measured during each internal probe acquisition. Hence the disparity in the amounts between internal and terminal probes. Importantly, our image analysis procedure alleviates possible effects of incomplete labeling by (i) referencing the probe signal to a signal in the PS shell (3’/5’) and by (ii) employing

particle averaging (**Figure 5C**). To clarify this, we added a short explanation on the averaging process (**results p.6§3**).

All probes have been designed using Stellaris Designer and the maximum number of probes received was used for the experiments. Although the efficiency of the individual FISH probes (521 different probes were used) is not known and difficult to determine, all probe sets provided a comparable signal intensity in the different *NEAT1* regions (**Figure S6 and S10; Table 4**). **Response Figure 4** shows individual raw images for all probe sets used in this study. The intensity distribution in these images do not hint at obvious signs of incomplete labeling. It is to be noted that lower signal to noise ratios would lead to a broadened intensity distribution with a lesser influence on the mean of the distribution.

We thank the reviewer for the suggestion to present representative actual images and added these to the manuscript (**Figure S11, results p.6 §3**)

We thank the reviewer for drawing our attention to the fact that selected examples in **Figure 5C** appear smaller. We indeed selected the wrong images and have corrected this error in the revised manuscript (correction in **Figure 5C**).

Response figure 4: Representative 2D cross-sections of individual spherical paraspeckles all 11 RNA-FISH probe sets. For visualisation all two-color 3D STED volumes were aligned rotationally and translationally using a transformation matrix calculated from Reference channel signal. Subsequently images are obtained by summing up the three centre XY layers. The top row shows fluorescence images of RNA-FISH probes hybridizing to the very 3' and 5' end of *NEAT1_2* (Reference; magenta). The middle row shows fluorescence images of RNA-FISH probes which each hybridize to the corresponding internal position of *NEAT1_2* (Internal; green). The bottom row shows the composite of Reference and Internal. Scale bar = 300 nm.

b) The authors stated that the signal of the 16-22K probe protrudes beyond the boundary delineated by the 5' and 3' ends, forming a small loop (Fig. 5D). This is an important discovery and further evidence is needed to convince the readers, e.g. clear demonstration of colocalization and co-polarization of this loop with the 3' end of *NEAT1_2* and/or protrusion beyond other surface markers of paraspeckles, Ideally.

Response: We are pleased that the reviewer highlights the importance of our discovery of an additional loop of *NEAT1_2* in paraspeckles in the region of about 16K to 22K nts, which also represents the largest distance to the paraspeckle center.

Our experiments were designed to rigorously map multiple distances along *NEAT1_2* with respect to a reference signal. As a reference signal, we chose the 5' and 3' ends of *NEAT1_2*, which define the shell region of paraspeckles^{3,4} and are qualified as surface markers.

Specifically, a small region at the very 5' (0 to 1K nts) and 3' end (22K to 22.7K nts) of *NEAT1_2* were chosen to act as a “reference” to describe the global size and shape of the paraspeckles. All of the other 11 probe sets stretching over the remaining ~93% of *NEAT1_2* dubbed “internal” were recorded together with the “reference” probes, and all distances were evaluated in comparison to the “reference”. Only the “internal” probes binding *NEAT1_2* in the regions from 18K to 20K protrude the shell boundary (defined by the “reference” signal) and excel the distances measured for all other “internal” regions. In addition, this finding is flanked by the distances measured for the adjacent regions 16-18K and 20-22K. Furthermore, the distribution shown in Figure 5D, again taking into account the reference signal, shows a continuous curve. Taken together, we are convinced that this proves beyond doubt the existence of the loop structure of *NEAT1_2* in the range 16K to 22K nts.

Following the suggestion of the reviewer, we added images of individual dual-labeled paraspeckles (**Response Figure 4** and **Figure S11**).

c) Data in Figure 5D should be carefully reviewed. The apparent discrepancy in the numbers of data points between the top and bottom 5th percentiles (probe) needs to be explained. For example, in the “5-7” group, only 23 data points fall within the 5th percentile, while more than 34 points are located in the bottom 5th percentile.

Response: The plot shown in **Figure 5D** is a zoom in and the remaining points are outside the range shown, which is why the number above and below may vary. The scale was chosen to direct the attention to the main part of the points and their median than to the outliers. After all, there are more than 690 data points per plot shown. We would like to thank the reviewer for noticing this and allowing us to maintain the highest scientific standards. For correction we amended **Figure 5D** by removing the outliers and adding a new panel **A** to **Figure S12** that includes all data points and proceeding the same way with the two plots in **Figure S12**. Captions of **Figure 5** and **S12** have been adapted accordingly.

Minor concerns:

i) Light purple color is used to indicate the major axes in shell-to-center distances (Fig. 3C)” and also the “ratio of major/minor axis (Fig. 3D)”. Consider using a different color in Fig. 3D for clarity.

Response: We thank the reviewer for his suggestion to enhance clarity. We changed the color in **Figure 3D** to black.

ii) Figure 4C and E show representative image of “44%<P<47%” and “35%<P<46%”, respectively. This is a slightly narrower bin than mentioned in the main text. Justification should be provided. Since the author assumes a certain distribution of the random degree of polarization, they should be able to provide the calculated significance of measured degree of polarization. Also, please clarify whether the scale bar is applied to the entire panel in Fig. 4C, 4D, and 5C.

Response: We thank the reviewer for drawing our attention to the inconsistency between the figures given in the main text and those in **Figure 4**. The numbers in the main text were incorrect and have been corrected (**results p.5 §4**).

Regarding the polarization, we have two interpretations of this comment and decided to address both:

(i) the distribution of polarization values was derived from analyzing the intensity distribution of 5' and 3' signals on the surface of single PS particles and is shown in **Figure 4BD**. There is no assumption on a random distribution underlying this analysis. However, and in response to a similar question raised by reviewer #3, we conducted a simulation of random distribution of 3' and 5' signals on the surface of PS particles (see **Response Figure 5**; and a detailed description on that simulation in the response to reviewer #3). Comparing this simulated distribution with the experimental distribution using a two-tailed Mann-Whitney-U-test yielded no significant difference in the populations ($p=0.59$).

Response figure 5: Compared distributions for polarization degree of 3' and 5' ends. Degree of polarization among spherical paraspeckles for experimental and simulated data.

(ii) Should the reviewer have referred to a random orientation (not polarization), it would be related to the data shown in Figure 4G. The random orientation of two vectors in 3D space is calculated as 1 radian $\approx 57^\circ$ and shown as a thin line in the graph. We added a short summary at the end of the paragraph that conveys the main message of the result. To enhance clarity, we have included a summarizing sentence at the end of this section (results p.6 §1).

The description of scale bars for the subpanels mentioned were clarified.

iii) Regarding the predicted NEAT1_2 orientation, the authors hypothesized that the 5' and 3' terminals of NEAT1_2 are distributed end to end radially from straight lines to "V" shapes. Response possibilities could also be consistent with their observation to explain the polarization, which should be discussed.

Response: Considering a minor contribution from NEAT1_1 (see above), and taking into account our experimental observations of non-polarised, partially polarised and fully polarised paraspeckles, more than one configuration must be possible. The very likely hypothesis is that next to V-shape, a straight-across configuration occurs. Yet we agree with the reviewer that

other configurations, of intermediate shape, might occur and for example explain partially polarized particles. For these reasons, and to leave room for other interpretations (which we do not have experimental access to yet, though), we phrased this rather broad statement in the discussion of the manuscript.

Reviewer #2 (Remarks to the Author):

1) The spatial resolution. The spatial resolution of the STED microscopy cannot be well defined by settings and optics. A characterization of the spatial resolution in 3D is recommended.

Response: We agree with the reviewer that a characterization of the spatial resolution should be given. Therefore we calculated the axial and lateral resolution of STED images acquired with Abberior STAR 580 (STAR 580) and Abberior STAR 635P (STAR 635P) using single image Fourier ring correlation¹². Results are shown in **Response Table 1** and are added to the manuscript (**Table 2; results p.4 §1**)

Response Table 1: Median Resolution of measured STED images. Resolutions were determined by performing single image Fourier ring correlation on 100 central cross sections (sum of 3 central slices) of images acquired using P3-STAR 580 and P3-STAR 635P. Errors are given as standard deviation.

Imager strand	Cross section view	Resolution / nm
P3-STAR 580	XZ	133±15
	XY	112±13
	YZ	123±15
P3-STAR 635P	XZ	109±13
	XY	100±16
	YZ	106±12

2) Bleaching effect. The same sample is imaged multiple times. The bleaching effect could be a problem, especially in the case of 3D imaging, where the step size is set to a very small value. Please comment on the bleaching effect that may affect the signal level and accuracy of the quantitative analysis.

Response: We agree with the reviewer that a bleaching effect may affect the results of the quantitative analysis. We didn't expect a significant bleaching effect influencing the presented data because of the following reasons: For both of the screening steps very low laser intensities were used (571 nm; 2.9 μ W at the back focal plane). This and the high amount of fluorophores per stained RNA region makes us very confident that there isn't any noticeable photobleaching occurring during the screening steps. For the 3D STED imaging we choose to record XZY stacks over XYZ due to recommendation^{13,14}. For 3D STED recording with a top-hat point spread function it is supposed to avoid a decreasing signal in later recorded sheets.

To check for bleaching, we studied the image intensities summed along the Y-direction, over the XZ planes. Since we recorded XZY stacks, bleaching would cause a sharp drop-off along the Y-direction. We compare this to a control of sums along the X-direction, with the ZY planes

summed. We found only a minor drop-off along the Y-direction (**Response Figure 6**). We can conclude that we did not experience detrimental bleaching.

Response Figure 6: Summed up intensities of recorded image planes along the X- e.g. Y-direction.

3) In Figure 2 iv), please indicate the meaning of the color

Response: In **Figure 2 vi)** the colors indicate the stained part of the exemplary paraspeckle. The shell region is indicated by magenta and the core region by green color. Clarification has been added to **Figure 2** caption.

4) In Paragraph 4 on Page 4, "...and plotted against particle size (Fig 3C) and as histograms (Fig S2)." Please confirm if it is the particle size in Fig. 3D.

Response: We thank the reviewer for spotting this. It is the particle number sorted for size as was done prior to the translational alignment and particle averaging. The sentence in the manuscript has been revised accordingly (**results p.4 §4**).

5) In Figure 3C, the x-axis is sorted bin number? Not image number? It seems the data point is for each bin, not for each image.

Response: The lines in Figure 3C indicate the envelope around the single image points. For a more clear figure, points were added that each correspond to the mean of 100 single images. Please note that these are not the same bins as were used for the translational alignment and particle averaging.

6) Figure 5C, please discuss why the internal probe images show bigger particles, especially at 1-3, 18-20, and 20-22.

Response: Reviewer #1 also raised a question very similar to this. Due to miscommunication on our part we selected the wrong images to present. More precisely, the averages for the reference probes were displayed with the wrong zoom settings. Therefore all reference particles appeared too small. The **Figure 5C** is now corrected in the manuscript.

Reviewer #3 (Remarks to the Author):

1. On page 13, you write that you projected the 3D intensity information on a spherical surface. Which projection did you use? It is not uniquely defined based on the information that you provide in the manuscript.

Two lines above equation 7 you mention that the spherical surface is expanded into spherical harmonics, but this is not true. The intensity function on the surface is expanded, not the surface itself.

Response: Thank you for bringing this detail to our attention, this is indeed what we meant. We added an explanation of the projection method and rewrote the paragraph for clarity (**methods p.15 §3**):

“We combined the aligned image pairs by subtracting the normalized 3' channel from the normalized 5' channel. We then projected the resulting 3D intensity information, in which negative values indicate 3' dominant regions and positive values 5' dominant regions, onto a spherical surface. To do so, the image volume was first divided into 512 spherical cones of equal volume. The intensity was summed in each cone. The spherical surface was then composed by attributing these sums to the surface points corresponding to the centres of the corresponding cones. Subsequently, the found intensity function on the spherical surface was expanded into spherical harmonics. The expansion of the intensity function $A(\Omega)=A(\theta,\phi)$ into spherical harmonics, with θ and ϕ the polar and azimuthal angles, respectively, is given by...”

2. In the line below equation 7 you need to present explicitly the equations for the spherical harmonics Y_l^m and expansion coefficients a_l^m as there exist different conventions for defining them. Without mentioning the equations that you used, the analysis cannot be reproduced.

Response: We included the definition in the main text (**methods p.15 §3**):

“The expansion of the intensity function $A(\Omega)=A(\theta,\phi)$ into spherical harmonics, with θ and ϕ the polar and azimuthal angles, respectively, is given by

$$A(\Omega) = \sum_{l=0}^{\infty} \sum_{m=-l}^l a_l^m Y_l^m(\Omega),$$

where Y_l^m is the spherical harmonic of degree l and order m , and a_l^m is the corresponding expansion coefficient. The spherical harmonic Y_l^m is given by Arfken et al., 2013. ¹⁵

$$Y_l^m(\theta, \varphi) = (-1)^m \sqrt{\frac{2l+1}{4\pi} \frac{(l-m)!}{(l+m)!}} P_l^m(\cos(\theta)) e^{im\varphi},$$

where P_l^m is the associated Legendre function. The expansion coefficients a_l^m are then found using:

$$a_l^m = \sum_{\varphi=\varphi_1}^{\varphi_N} \Delta\varphi \sum_{\theta=\theta_1}^{\theta_N} \Delta\theta Y_l^m(\theta, \varphi)^* A(\theta, \varphi).$$

The summation reflects the discrete nature of the computational calculations.”

3. There are several problems with equation 8.

First, "relative weights of the first degree ... expansion coefficients" sounds a bit unspecific. For example, it does not suggest that the coefficients are squared in equation 8.

Response: Thank you for addressing this. We have updated the manuscript to be more explicit about our measure of the degree of polarisation (**methods p.15-16**). See the discussion below for more details on this measure.

Second, I see no justification for defining the polarization as in equation 8. Here the polarization can even diverge. Why did you not simply calculate the angular power spectrum of the intensity distribution and choose the degree $l=1$ contribution from the angular power spectrum (or even better its square root, as it is frequently done in the literature) as a value for the polarization? The angular power spectrum is widely used and well known and would be a better choice than your definition.

Third, instead of dividing by the sum of all squared expansion coefficients in equation 8 (which is wrong, if I remember correctly, if one does not ensure equal contributions of the different degrees by including appropriate prefactors), you should simply normalize the intensity distribution function before you start to expand it.

Fourth, you should have taken the absolute squares of the coefficients and not just their squares. These coefficients have complex values. This means that with your definition of equation 8 you get complex values for the polarization, which does not make sense and is not in line with what you present in the main text.

Response: We thank the reviewer for noticing the typo in the original equation regarding the absolute values. We have corrected this (**methods p.16 eq.12**). We hereby expand on our choice of measure of the degree of polarisation.

We used the spherical harmonics Y_l^m in their orthonormal definition (eq. 8), with

$$\int Y_{l_1}^{m_1} Y_{l_2}^{*m_2} d\Omega = \delta_{l_1 l_2} \delta_{m_1 m_2}.$$

The orthonormality is useful when expressing the power spectrum of the intensity distribution on the spherical surface, $A(\Omega)$, using the spherical harmonics expansion:

$$\text{Power} = \int |A(\Omega)|^2 d\Omega = \int \left| \sum_{l=0}^{\infty} \sum_{m=-l}^l a_l^m Y_l^m(\Omega) \right|^2 d\Omega = \sum_{l=0}^{\infty} \sum_{m=-l}^l |a_l^m|^2.$$

Hence, the contribution of each degree to the power spectrum is given by the sum of the squared moduli of the expansion coefficients:

$$C_l = \sum_{m=-l}^l |a_l^m|^2.$$

We are interested in the relative contribution of the first-degree spherical harmonics. To measure the degree of polarisation, we normalised the contribution of these first-degree spherical harmonics to the total power:

$$P = \frac{C_1}{\sum_{l=0}^{\infty} C_l}.$$

This degree of polarisation takes values between 0 (no polarisation) and 1 (fully polarised). It does not diverge, as C_1 is included in the normalisation sum. We have added the four expressions above to the manuscript (**methods p.16 eq.10-13**). We also updated **Figure 4A** and the corresponding caption to include equation 13, the detailed expression for the degree of polarization.

To allow for intuitive interpretation of this measure, it should follow roughly a normal distribution between 0 and 1, centred at 0.5, for a completely random distribution of the 5' and 3' ends of *NEAT1_2* across the paraspeckles surface. To verify that our measure fulfills this, we simulated 1000 such paraspeckles and found the following distribution (with the degree multiplied by 100 to get a percentage). Histogram and representative examples are in **Response Figure 7**. Of note, this is in line with the findings from our experimental data (**Figure 4B, C**). We can conclude that our measure for the degree of polarisation accurately represents the contribution of the first-degree spherical harmonics to the power spectrum of our signal. Furthermore, it follows an approximately random normal distribution between 0 and 1, with a

mean around 0.5, for a randomly distributed signal. We added the simulated data of spherical paraspeckles to the manuscript (**Figure S7; results p.5 §4; methods p.16 §5**).

Response Figure 7: Polarisation degrees for simulated paraspeckles with random distributed 3' and 5' ends of *NEAT1_2*. **A** Distribution of measured degree of polarization for 1000 simulated paraspeckles. **B** Examples of simulated spherical paraspeckles with the 5' and 3' ends of *NEAT1_2* randomly distributed across the paraspeckle shell region (**Methods**).

- Before equation 9 you cite reference 45. Which reference do you mean here? The list of references for the manuscript ends at reference 40.

Response: We thank the reviewer for noticing this issue with a few citations. We have checked the references and corrected the errors.

- Did you ensure that reference 45 uses the same convention for the spherical harmonics and corresponding expansion as you? If you used not the same convention, equations 9-11 are wrong in your case.

Response: The same convention was used as in the reference.

- Equation 9 is not needed at all. Instead of calculating first a spherical harmonics expansion and using then a representation of the polarization vector in spherical harmonics, you could simply expand the known intensity function into a Cartesian expansion up to degree 1. The degree 1 contribution is the polarization vector.

Response: There are indeed different ways to estimate the polarisation vector. We have already calculated the spherical harmonics expansion to find the degree of polarisation. We use the same expansion to find the polarisation vector to avoid doing additional calculations.

7. Below equation 12 you mention a Fibonacci grid. There is no point in using a Fibonacci grid here (and the restriction to the polar angle seems to make is even wrong). When you want to have an expression for the angle between the z axis and a random point on the sphere, you can simply write down the unit orientation vector in 3D parametrized by spherical coordinates, multiply it by the unit vector along the z axis (scalar product), and calculate the arccos from it to obtain the angle between the z axis and another point on the sphere. Afterwards, you just need to average this over the spherical coordinates by a surface integral with a normalizing prefactor $1/(4\pi)$.

Response: Following the proposed method (but integrating over a hemisphere, as the polarisation vector falls between 0 and $\pi/2$):

$$\arccos(\hat{n} \cdot \hat{z}) = \arccos([\sin(\theta) \cos(\varphi), \sin(\theta) \sin(\varphi), \cos(\theta)] \cdot [0,0,1]) = \arccos(\cos(\theta)) = \theta$$

$$\beta = \frac{1}{2\pi} \int_0^{2\pi} \int_0^{\pi/2} \arccos(\hat{n} \cdot \hat{z}) \sin(\theta) d\theta d\varphi = \frac{1}{2\pi} \int_0^{2\pi} \int_0^{\pi/2} \theta \sin(\theta) d\theta d\varphi = 1 \text{ rad} \approx 57.3^\circ.$$

Of course, the result is the same as with the Fibonacci grid. We have replaced the Fibonacci grid method with the suggested derivation for simplicity (**methods p.17 eq.18-19**).

Due to these issues I cannot recommend publication of this article in its present form.

Reviewer #3 (Remarks on code availability):

As my report mentions some issues with the analysis, the code needs to be revised accordingly.

We have updated the code according to the discussion above.

References

1. Chujo, T. *et al.* Unusual semi-extractability as a hallmark of nuclear body-associated architectural noncoding RNAs. *The EMBO Journal* (2017)
2. Li, R., Harvey, A. R., Hodgetts, S. I. & Fox, A. H. Functional dissection of NEAT1 using genome editing reveals substantial localization of the NEAT1_1 isoform outside paraspeckles. *RNA* **23**, 872–881 (2017).
3. Souquere, S., Beauclair, G., Harper, F., Fox, A. & Pierron, G. Highly ordered spatial organization of the structural long noncoding NEAT1 RNAs within paraspeckle nuclear bodies. *Mol Biol Cell* **21**, 4020–4027 (2010).
4. West, J. A. *et al.* Structural, super-resolution microscopy analysis of paraspeckle nuclear body organization. *J. Cell Biol.* **214**, 817–830 (2016).
5. McCluggage, F. & Fox, A. H. Paraspeckle nuclear condensates: Global sensors of cell stress? *BioEssays* **43**, 2000245 (2021).
6. Unveiling the intricacies of paraspeckle formation and function. *Current Opinion in Cell Biology* **90**, 102399 (2024).
7. Naveed, A., Fortini, E., Li, R. & Fox, A. H. Long Non-coding RNAs and Nuclear Body Formation and Function. *Molecular Biology of Long Non-coding RNAs* 65–84 (2019).
8. Nakagawa, S., Yamazaki, T., Mannen, T. & Hirose, T. ArcRNAs and the formation of nuclear bodies. *Mamm Genome* **33**, 382–401 (2022).
9. Hirose, T., Ninomiya, K., Nakagawa, S. & Yamazaki, T. A guide to membraneless organelles and their various roles in gene regulation. *Nat Rev Mol Cell Biol* **24**, 288–304 (2023).
10. Hellmeier, J. *et al.* Quantification of absolute labeling efficiency at the single-protein level. *Nature Methods* **21**, 1702–1707 (2024).
11. Hirose, T. *et al.* NEAT1 long noncoding RNA regulates transcription via protein sequestration within subnuclear bodies. *Mol Biol Cell* **25**, 169–183 (2014).
12. Rieger, B., Droste, I., Gerritsma, F., Ten Brink, T. & Stallinga, S. Single image Fourier

- ring correlation. *Opt. Express* **32**, 21767–21782 (2024).
13. Velicky, P. *et al.* Dense 4D nanoscale reconstruction of living brain tissue. *Nature Methods* **20**, 1256–1265 (2023).
 14. Mol, F. N. & Vlijm, R. Automated STED nanoscopy for high-throughput imaging of cellular structures. *bioRxiv* 2022.09.29.510126 (2022) doi:10.1101/2022.09.29.510126.
 15. Arfken, G. B., Weber, H. J. & Harris, F. E. *Mathematical Methods for Physicists A Comprehensive Guide*. 715–772 (Academic Press, 2013).

Response to Reviewer comments

We thank all the reviewers for investing time in assessing our revised manuscript and for the valuable feedback. We have addressed all points raised by the reviewers. As part of this revision, new data was generated, analysed and interpreted.

Reviewer #1 (Remarks to the Author):

The authors have made extensive efforts to address the questions raised in the last round of review and indeed answered/explained many of the questions raised in the rebuttal and the text. However, there are still substantial concerns that remain and need to be further addressed.

1. The authors cited previous reports and argued that HeLa cells harbor NEAT1_1:NEAT1_2 at 1:9 ratio, thus the common 5' probe used in this manuscript "should not interfere with the results". However, these previous studies still do not exclude NEAT1_1 in paraspeckles (PSP). In fact, it is unknown whether 1:9 ratio of NEAT1_1/NEAT1_2 from total RNA on northern indeed represents the ratio of these two NEAT1 isoforms in PSP. The new data in Response Figure 1 used a probe for the NEAT1_2-specific 5-7 kb region in a polarized PSP example, which is localized more in the center relevant to the reference 5'-3' probe. Nonetheless, this still does not definitively address the 5'-3' probes are indeed from NEAT1_2. Whether NEAT1_1 abundance could corrupt the 'reference' signal, thereby compromising the fundamental interpretation of PSP outlines, is critical for NEAT1/PSP field, which should be clearly demonstrated. The ideal experiment is to eliminate NEAT1_1, or at least apply this method to U2OS cells that express abundant NEAT1_1 and show whether NEAT1_1 abundance can interfere with the analysis/conclusion. This will address whether the method is limited only to cells have negligible NEAT1_1 but could not be applied to many other cells that express abundant NEAT1_1. If this is the case, the limitation of using this method for studying paraspeckles in cells with abundant NEAT1_1, should be clearly indicated.

Response: We thank the reviewer for this important comment. We understand the concern that, albeit that the *NEAT1_1* content is very low compared to *NEAT1_2* in HeLa cells (1:9 ratio), we did not confirm that this ratio holds also true for the paraspeckles. We thank the reviewer for suggesting an experiment to address this concern, which we have carried out as part of this revision and whose results we discuss below (**Response Figure 8**). We will begin this response with a short summary on the current literature, which will be helpful as background information and supports our rationale.

Sasaki et al. reported that *NEAT1* is uniquely enriched in the high-density nucleoplasmic fraction (Sasaki et al. 2009). Nishimoto et al. reported that >93% and >96% of all *NEAT1_2* foci overlapped with the paraspeckle proteins TDP-43 and FUS/TLR, respectively, indicating that almost all *NEAT1_2* is localized inside paraspeckles (Nishimoto et al. 2013). In contrast to that, *NEAT1_1* was reported to accumulate in paraspeckle-independent speckles (microspeckles) (Li et al. 2017). Taken together, the fraction of *NEAT1_2* in paraspeckles might even be higher than the ratio to *NEAT1_1* suggests (of 9:1 in HeLa cells; see

Response Figure 9) (Chujo et al. 2017). We agree with the reviewer that the northern blot data does not necessarily reflect the *NEAT1_2/NEAT1_1* content in paraspeckles.

We followed the suggestion of the reviewer and performed new experiments as part of this revision. We measured paraspeckles in U-2 OS cells using two sets of FISH probes: the first probes bind to the 5' end (0-1K nt) of *NEAT1* (targeting both *NEAT1_2* and *NEAT1_1*), and the second set to the 3' end (22-22.7K nt) targeting only *NEAT1_2*. **Response Figure 8A** shows confocal images of both U-2 OS and HeLa cells, where the *NEAT1_2* foci (identified via 3' end signal) are found overlapping with signal originating from the 5' end probe for both cell lines. However, U-2 OS cells exhibited fewer *NEAT1_2* foci (~ 3 times less), and the area of these foci was smaller (~1.5 times smaller) than in their HeLa counterparts (**Response Figure 8B, C**). This is in line with the previous observation of lower *NEAT1_2* expression levels in U-2 OS cells than in HeLa cells (Chujo et al. 2017) (**Response Figure 9**). Importantly, the mean fluorescence intensity for both 3' end and 5' end probes showed no significant difference in both cell lines (**Response Figure 8D**). In addition, no structural differences were observed in STED images of paraspeckles (**Response Figure 8E**). Mean intensity analysis of the higher-resolution STED data of both 3' and 5' ends also showed no significant difference (**Response Figure 8F**). In conclusion, the observation of no apparent differences in the *NEAT1_1/NEAT1_2* intensities in paraspeckles indicates that the composition of paraspeckles in U-2 OS and HeLa cells is very similar, despite the different expression ratio of the two isoforms. This is in accordance with literature reporting that *NEAT1_1* has a PS-independent role (Li et al. 2017).

The above data of 5'/3' intensities in HeLa and U-2 OS indicates that *NEAT1_1* is not a structural component of paraspeckles. In conclusion, we can safely assume that the 5' reference signal is not corrupted in paraspeckles, and that our structural analysis hence is valid. Moreover, this hints at a general building principle of paraspeckles. However, we agree with the reviewer that for such a statement, it would require a more thorough examination. Interestingly, for both U-2 OS and HeLa cells, we observe a similar background signal all over the nucleoplasm (excluding PS), which might reflect the *NEAT1_1* localization in other regions in the nucleus. However, that will require future studies.

We made according changes in the manuscript and now (1) emphasize that our study targeted paraspeckles in HeLa cells, and (2) discuss the possible influence of isoform expression level in context with the current literature. We have not included the additional data in the manuscript, since we believe that this requires an extensive and biology-focused thorough examination, which would merit a study on its own.

Response Figure 8: Comparison of *NEAT1_1* and *NEAT1_2* in HeLa and U-2 OS cells. A Confocal images of HeLa and U-2 OS cells labeled with FISH probes targeting the 5' end of *NEAT1* (0-1K nt; binding both isoforms) and the 3' end (22-22.7K nt; binding *NEAT1_2* exclusively) (scale bar = 20 μm). **B** Number of *NEAT1_2* foci detected per cell. Bars and whiskers indicate mean and s.e.m. Significance was tested using a two-tailed student's t-test. **: $p < 0.01$, $N = 201$ (HeLa) and $N = 156$ (U-2 OS). **C** Mean size (area) of *NEAT1_2* foci in HeLa and U-2 OS cells. Black dotted line represents the median. Significance was tested using a Mann-Whitney-U-test. ****: $p < 0.0001$, $N = 796$ (HeLa), N

= 201 (U-2 OS). **D** Mean fluorescence intensity of the 5' and 3' ends of *NEAT1_2* foci in HeLa and U-2 OS cells (confocal data). Black dotted line represents the median. Significance was tested using a two-tailed student's t-test (for 5' end distributions) and a Mann-Whitney-U-test (for 3' end distributions). ns: $p > 0.05$. $N = 225$ (HeLa), $N = 145$ (U-2 OS). **E** STED images of HeLa and U-2 OS cells labeled at the 5' and 3' ends (scale bar = 300 nm). **F** Mean fluorescence intensity of the 5' and 3' ends of *NEAT1_2* foci in HeLa and U-2 OS cells (STED). Black dotted line represents the median. Significance was tested using a two-tailed student's t-test. ns: $p > 0.05$. $N = 84$ (HeLa), $N = 73$ (U-2 OS).

Response Figure 9: Quantitative characterization of NEAT1 (Figure and caption taken from Chujo et. al, 2017, original Figure 2DE (Chujo et al. 2017)). **Left:** “Steady-state levels of NEAT1_2 and NEAT1_1 in various human cell lines. An example of RPA using total RNA extracted with the improved extraction method is shown in the upper panel. In the lower panel, NEAT1 levels (expressed as relative molarities) are shown relative to NEAT1_2 level in HeLa cells.”; **Right:** “RNA-FISH of NEAT1_2 (green) in various human cell lines. Nuclei were stained with DAPI (blue).”

2. Response Fig.2/S2: The text indicates 93% whereas the rebuttal indicates 95% of PSP contains PSPC1. Such colocalization is convincing but which is correct? PSPC1 appears to be localized in the center of PSPs (panel A). What is the shell-to-center distances of PSPC1 compared to the 3'-5' probe at the surface?

Response: We apologize for the discrepancy and appreciate the reviewer for pointing this out. The correct number is ~95 %. The manuscript has been updated accordingly (p4 §1).

PSPC1 is a PSP that is known to be located in the centre of PS (Hao et al. 2025), and the reviewer is fully correct. Results of the shell-to-centre distance ratio analysis, presented in the submitted manuscript, are plotted in **Response Figure 10** together with data from different *NEAT1_2* regions for comparison. We note that imaging data of PSPC1 and the *NEAT1* shell data was recorded [as part of the first revision] to demonstrate the robustness of the particle selection procedure, yet not with acquisition settings required for high-resolution quantitative distance analysis: the sample size is smaller ($n(\text{PSPC1}) \sim 200$; $n(\text{internal FISH})$

probes) ~ 1000), and the spatial resolution is different. This allows for a qualitative comparison (**Response Figure 10**). A fully quantitative comparison would require a higher number of particles and isotropic 3D STED imaging settings. Therefore, we did not include this data in the manuscript. Nevertheless, we agree with the reviewer that the investigation of the position of proteins within paraspeckles and relative to *NEAT1_2* is an exciting opportunity for future research.

Response Figure 10: Average localization of *NEAT1_2* regions and PSPC1 inside the paraspeckle. The average distance of the 4 different *NEAT1_2* regions and PSPC1 to the paraspeckle center, relative to the to-center distance of the combined 5'/3' reference positions, as calculated from radial intensity distribution. Lines inside the boxes indicate median, the box indicates the 25th and 75th percentile and the whiskers indicate the 5th and 95th percentile.

3. The MG132 data is a direct test for the method. This reviewer suggests including this data in supplemental info and discusses the results relevant to the literature of PSP elongation upon exposure to MG132.

Response: We thank the reviewer for the suggestion. Following the advice of the reviewer, we have included the MG132 data in the revised manuscript and discussed its implications (**Fig S5B, C; results p.5 §1; discussion p.9 §2; methods p.10 §2**).

4. Response Fig. 4 nicely showed 18K to 20K probe protrudes the shell boundary. Does this protrusion exhibit co-polarization with the 3' end of *NEAT1_2*? If such co-polarization is not consistently observed, would it undermine the interpretation?

Response: We thank the reviewer for this question. To verify whether the protruding loop co-localizes/co-polarizes with the 3' end, we labeled paraspeckles in HeLa cells using two FISH probes, a first one targeting the 3' loop region (18-20K nt) and a second one targeting the 3' end (22-22.7K nt), and conducted STED microscopy experiments (**Response Figure 11A**). To answer the question of the reviewer, we particularly selected paraspeckles that showed an incomplete ring in the 3' end channel, which we interpreted as polarized paraspeckles (**Response Figure 11B**). In these paraspeckles, we observe an overlap of the

3' end and 3' loop regions, indicating close proximity. This overlap might be better described as 'co-arrangement': since the 3' loop is located outside the shell, the overlap to the 3' end appears not complete; however, the spatial patterns are almost identical (an inflation of the images of the 3' end would lead to almost perfect overlap with the images of the 3' loop).

To better contextualise the observation, we compared the 3' end/loop region data with paraspeckles with 5' end/3' end staining (see **Response Figure 8E**, left panel): those paraspeckles which show an incomplete ring in the 3' end signal show a 5' end signal that complements the structure of the ring (polar, **Response Figure 11C**). This supports the hypothesis that in paraspeckles exhibiting a polarized distribution of the 5' and 3' ends, the 3' end is in close proximity to the loop region. In addition, we calculated the Pearson correlation coefficient for both 3' end measured together with loop region (0.67 ± 0.15) and 3' end measured together with the 5' end (0.51 ± 0.17) which supports the visual inspection (**Response Figure 11D**).

We include this data in the revised manuscript (**Figure S13; results p.7 §1; methods p.11 §5, p.17 §5**).

Response Figure 11: Two-color STED experiments of paraspeckles labeled at the 3' loop and 3' end, and at the 3' and 5' ends, of *NEAT1_2*. **A** Co-staining of the protruding 3' loop region (18-20K nt) and the 3' end (22-22.7K nt) (scale bar = 300 nm). **B** Selected images showing potentially polarized paraspeckles. FISH probes hybridized in the protruding loop region (18-20K nt) and positioned directly at the 3' end (22-22.7K nt) (scale bar = 300 nm). **C** Selected images showing potentially polarized paraspeckles labeled at the 5' end (0-1K nt) and the 3' end (22-22.7K nt) (scale bar = 300 nm). **D** Pearson correlation coefficient for paraspeckles labeled at the 3' end and 3' loop, and for paraspeckles labeled at the 3' and 5' ends. Significance was tested using a two-tailed student's t-test (****: $p < 0.0001$). $N = 199$ (3' end and 3' loop), $N = 208$ (3' end and 5' end).

5. According to the comparison against simulated random distribution in response figure 5, there is no significant polarization of either 5' or 3' end of NEAT1 on paraspeckles. Nonetheless, the subsequent analysis is a major claim. It is unclear why to spend large efforts to chase this statistically insignificant phenomenon? Also, the authors should run a similar simulation on elliptical particles to address that the correlation between polarizing angle and major axis is not an artifact from categorizing algorithms.

Response: We thank the reviewer for this comment. The key finding of this analysis is that paraspeckles are not built in an ordered way, but rather all possible orientations of *NEAT1_2* are observed, that is, all possible configurations that result from a random distribution.

Paraspeckles were predominantly thought to be built with a V-shape arrangement of *NEAT1_2* (McCluggage and Fox 2021; Ingram and Fox 2024; Naveed et al. 2019; Nakagawa et al. 2022; Hirose et al. 2023). From our observations of highly or moderately polarized paraspeckles that were not reported before (**Figure 4C,E**), we concluded that the V-shaped model cannot be correct for a large number of paraspeckles, as this model would only allow for a homogenous distribution (non-polar) of the signal from both RNA ends across the paraspeckle surface. The presence of highly polarised paraspeckles necessitates the possibility of the two *NEAT1_2* ends to adopt different orientations, including the possibility that they are arranged in a straight conformation through the paraspeckle.

In order to support this interpretation of our data, we performed simulations where the 5' and 3' ends were free to distribute on a spherical surface (a random distribution of ends, rather than a V-shape distribution). These simulations resulted in particles with signal distributions ranging from “non-polar” to “polar” (**Figure S7B**) and were similar in their occurrence to experimentally found structures of paraspeckles. Moreover, the histogram of all possible polarization arrangements of simulated particles matched the data of experimentally measured polarization in paraspeckles very well (**Figure 4B, S7A**). Taken together, our data and simulation indicate that *NEAT1_2* 5' and 3' ends arrange not only into a V-shape, but also into other configurations including those where the RNA spans the entire length of the minor axis of the paraspeckle.

To address the concern that the relationship between the polarizing angle and the major axis is an algorithmic artefact, we extended our simulation and now include ellipsoidal paraspeckles with randomly distributed 5' and 3' ends over its surface. **Response Figure 12** shows four boxplots, corresponding to the real/simulated and spherical/elongated paraspeckles, showing the angles between the polarization axis and the elongation axis for polarized particles (polarization > 75%). This analysis shows that the angle between the polarisation vector and the elongation axis has a broad range around the random angle of ~57 degrees (1 rad) in the case of real spherical, simulated spherical and simulated elongated paraspeckles; only the real elongated paraspeckles show a more narrow distribution around a non-random angle. This allows us to conclude that the relationship found between the polarisation and the elongation in the real polarised elongated paraspeckles is not an analysis artefact. On that note, our results also show that elongated paraspeckles exhibit a polarization along the long axis. This hints that this observation might be relevant for growth and division, which would be interesting to address in the future.

Response Figure 12: Angle of polarization vector with elongation axis for real and simulated paraspeckles. Shown is the angle that was calculated between the polarisation vector and the vector pointing in the direction of paraspeckle elongation for real and simulated spherical ((ellipticity > 1.3)) paraspeckles as well as for real and simulated elongated (ellipticity > 1.3). The red line indicates the median, box indicates the 25th and 75th percentile and the whiskers indicate the 5th and 95th percentile. Black line indicates the random angle of ~57 degrees.

6. Nearly all original equations are improperly rendered after the revision (P11-16). Please double check and correct.

Response: We apologize for this mistake, which was a consequence of incorrect conversion into the .pdf format.

Reviewer #2 (Remarks to the Author):

The authors have addressed all my questions very well. I have no further questions.

Response: We thank the reviewer for the time and valuable feedback throughout this revision.

Reviewer #2 (Remarks on code):

The code is published on GitHub with a well-written README file. I am glad the authors mentioned the version of DIPLIB because the current version need to be built from source code. DIPLIB 2.9 can be installed with an executable installation file. However, I was not able to run the demo because of following error message in MATLAB (ver. 2023a):

```
Unrecognized function or variable 'PgetLocs'.
Error in PRotation (line 43)
[locs, Nlocs, leg] = PgetLocs([projDir '\ headDir]);
```

Nonetheless, the documentation is very clear and useful for setting up the code.

Response: We thank the reviewer for testing the software. The presented error likely arose due to an incomplete path. To avoid this issue in the future, the software has been modified accordingly (path set in the script instead of manually), such that this error should not occur any longer.

Reviewer #3 (Remarks to the Author):

The authors have strongly improved the presentation of their methodology. The mentioned errors have been corrected. With the added descriptions the previous questions have been answered and the previous confusion has been removed.

Response: We thank the reviewer for the time and valuable feedback throughout this revision.

I have only a few further remarks. When these issues get solved, I no longer have any objections to publication of this work.

- Above Eq. (7), "...the image volume was first divided into 512 spherical cones of equal volume.": Using spherical cones it is not possible to divide a volume into pieces without gaps or overlaps. There exist better methods that allow to partition a volume into solid angles without gaps and overlaps. This is equivalent to partitioning the surface of a sphere into areas without gaps and overlaps. The parts might then have different volumes, which is not a problem, but requires to be taken into account by appropriate weight factors.

Response: We thank the reviewer for the opportunity to motivate our projection method. We indeed chose to prioritise equal volumes contributing to equal surface areas, allowing us to attribute equal image volumes to all 512 equidistant points on the surface. We defined the cones such that their surface areas (spherical caps) add up to the total spherical surface area. As noted by the reviewer, this came at the cost of gaps and overlaps between the surface elements (spherical caps).

We can calculate the overlaps between all 512 spherical caps using an analytical method [<https://math.stackexchange.com/q/4028073>]

$$\begin{aligned}A_o &= 2r^2(\pi - 2A \cos(\theta_c) - B) \\A &= 2 \arctan\left(\frac{k}{\sin(s - \theta_c)}\right) \\B &= 2 \arctan\left(\frac{k}{\sin(s - \theta_d)}\right) \\k &= \sqrt{\frac{2\sin(s - \theta_c) \sin(s - \theta_d)}{\sin(s)}} \\s &= \frac{1}{2} (2\theta_c + \theta_d)\end{aligned}$$

where A_o is the area of overlap between the two spherical caps, θ_c is the half angle of the spherical caps (in our case, 5 degrees for all cones) and θ_d is the angle between the caps (value dependent on the pair of cones).

Performing this calculation for all 512 spherical caps, we find an average overlap area of 8.7%, which is 70 nm² per spherical cap. Considering our pixel area of 30x30 = 900 nm² and resolution of 100-120 nm, we consider this an acceptable error.

To verify that our chosen method does not qualitatively affect our results, we repeated the polarisation analysis using a different volume-to-surface projection method. In this alternative method, we again distribute 512 points evenly across a spherical surface. We use these points to construct a 3D grid in spherical coordinate space, with 6 radial sections as the third dimension. We then resample our images in this 3D grid, allowing us to sum over the radial dimension to get the summed intensity per surface element. With this method, we no longer have overlaps or gaps between the sections.

Below we present the resulting degrees of polarisation for four cases: imaged spherical and ellipsoidal paraspeckles analysed with our existing projection method and with the alternative projection method described above.

We observe that there is no qualitative difference between the results from the two projection methods. We therefore have chosen to keep our existing figures and results. We did update the code to include the alternative projection method.

- Eq. (9): Introduce the symbols $\Delta\varphi$, $\Delta\theta$, φ_1 , φ_N , θ_1 , θ_N

Response: We thank the reviewer for noting these missing symbol introductions. We have updated the manuscript to include them (**methods p. 16 § 1, equation 9**).

- Above Eq. (19): replace "integrated" by "averaged" as you not only integrate, but also divide by 2π .

Response: We thank the reviewer for noting this oversight, this is indeed what we meant. We have updated the manuscript to correct this.

Reviewer #3 (Remarks on code):

The code contains a README file, but I did not find sufficient information there on how to install or run the code.

Response: We thank the reviewer for raising this point. The README file referred to the demo manual included in the demo folder. We have now copied the information from the demo manual to the README file.

Editorial comments:

To maximise the reproducibility of research data, we strongly encourage you to provide a file containing the raw data underlying the following types of display items.

Response: A source data file containing all data underlying the display items in the manuscript has been submitted with this revision.

Please replace your bar graphs with plots that feature information about the distribution of the underlying data. All data points should be shown for plots with a sample size less than 10. For larger sample sizes, please consider box-and-whisker or violin plots as alternatives. Measures of centrality, dispersion and/or error bars should be plotted and described in the figure legend.

Response: We followed this instruction and made changes accordingly. We note that we keep the bar plot in **Figure S2B**, as it visualizes a single number value (number of particles colocalizing the protein PSPC1 and the structural RNA *NEAT1*) rather than a distribution.

Response to Reviewer comments

We thank all reviewers for assessing the revised manuscript and for the valuable feedback. We have addressed all points raised by Reviewer #3. As part of this revision, the methods section of the manuscript has been updated.

Reviewer #3 (Remarks to the Author):

1) Above Eq. (7), "...the image volume was first divided into 512 spherical cones of equal volume.":

The authors have explained in their rebuttal why they want to keep their method of dividing the volume into cones and although it is not the best possible method I agree that choosing a better method would not change the results to a relevant extent and is therefore not necessary. However, the authors did not change their manuscript in any way that prevents other readers from raising the same criticism as I did when I read this work. In particular, the authors should mention in their manuscript the following:

- The method that is used to divide the image volume into cones*
- The diameter (or solid angle) of each cone*
- That there are overlaps between neighboring cones*
- Some numbers that show the readers that using these cones is not a problem:*
 - Give an upper bound for the percentage of the overlap volume of all cones compared to the total volume of all cones.*
 - Give an upper bound for the percentage of the image volume that does not belong to any cone compared to the total image volume.*

Response: We thank the reviewer for pointing at this missing information. Following these suggestions, we have updated the manuscript [Methods, p. 16 §5].

2) In the line below Eq. (9) it should read "azimuthal" instead of "azimuth".

Response: We have corrected the manuscript accordingly [Methods p. 17 §3].